# UNAFLOW: a holistic wind-tunnel experiment about the aerodynamic response of floating wind turbines under imposed surge motion

Alessandro Fontanella[1], Ilmas Bayati[2], Robert Mikkelsen[3], Marco Belloli[1], and Alberto Zasso[1]

[1]Mechanical Engineering Department, Politecnico di Milano, Milano, Via La Masa 1, 20156, Italy.
[2]Maritime Research Institute Netherlands (MARIN), Wageningen, 6708 PM, The Netherlands.
[3]Technical University of Denmark (DTU), Department of Wind Energy, Lyngby, Denmark.

**Correspondence:** Alessandro Fontanella (alessandro.fontanella@polimi.it)

**Abstract.** Floating offshore wind turbines are subjected to large motions due to the additional degrees of freedom of the floating foundation. The turbine rotor often operates in highly-dynamic inflow conditions and this has a significant effect on the overall aerodynamic response and turbine wake. Experiments are needed to get a deeper understanding of unsteady aerodynamics, and hence leverage this knowledge to develop better models, and to produce data for the validation and calibration of existing numerical tools. In this context, this paper presents a wind-tunnel experiment about the unsteady aerodynamics of a floating turbine subjected to surge motion. The experiment results cover blade forces, rotor-integral forces and wake. 2D sectional model tests were carried out to characterize the aerodynamic coefficients of a low-Reynolds airfoil with harmonic variation of the angle of attack. The lift coefficient shows a hysteresis cycle close to stall, which grows in strength and extends in the linear region for motion frequencies higher than those typical of surge motion. Knowledge about the airfoil aerodynamic response was utilized to define the wind and surge-motion conditions of the full-turbine experiment. The turbine global aerodynamic response is evaluated from rotor-thrust force measurements, because thrust influences the along-wind response the floating turbine. It is found experimental data follow reasonably well predictions of quasi-steady theory for reduced frequency up to 0.5. For higher surge-motion frequencies, unsteady effects may be present. The turbine near-wake was investigated by means of hot-wire measurements. The wake energy is increased at the surge frequency, and the increment is proportional to the maximum surge velocity. A spatial analysis shows the wake energy increment is in correspondence of blade-tip and of the most loaded rotor region. PIV was utilized to visualize the blade-tip vortex and it is seen the vortex travel speed is modified in presence of surge motion.

## 1 Introduction

Floating offshore wind is receiving growing interest as it enables deep sea wind energy resource to be harvested at a competitive price, which is not possible with conventional bottom-fixed solutions. Floating offshore wind turbines (FOWTs) are subjected to large and low-frequency motions that may cause unsteady aerodynamics effects. The rotor of an FOWT operates in dynamic inflow conditions and, as pointed out by de Vaal et al. (2014), this occurs for two main reasons: platform motion modifies the

wind speed seen by rotor and in some cases moves the rotor in its own wake. This has a significant effect on the aerodynamic loads and, consequently, on the FOWT response. Moreover, platform motions result in large-scale movement of the turbine wake, which is relevant for the wake-interaction problem in floating farms (Wise and Bachynski (2020)).

Wind turbines and wind farms are often designed by means of engineering tools that were adapted from land-based tools. In this adaptation process, aerodynamic models have remained almost unchanged. However, floating turbines are subjected to peculiar inflow conditions that are not present in land-based turbines. The rotor of the latter undergoes small-amplitude motions associated to tower flexible response. The motion of an FOWT rotor is in large part set by the rigid-body motion of the support platform and is in general of higher amplitude and lower frequency than in land-based turbines. The accuracy of land-based-derived aerodynamic tools in this new inflow conditions is yet to be assessed. An accurate prediction of the aerodynamic response caused by rotor motion is crucial. As mentioned, this occurs at lower frequencies than in land-based turbines and, unlike the latter, causes significant interactions with the turbine controller (i.e., the aerodynamic response in FOWTs is inside the bandwidth of the turbine controller). Experiments play a crucial role in verifying whether aerodynamic codes are accurate also for floating turbines, to get a deeper understanding of the aerodynamic phenomena that occurs when the wind turbine undergoes large motions and, based on this knowledge, to develop better simulations tools.

To date, there are few wind-tunnel experiments that shed light into the unsteady aerodynamic response of floating turbines. Farrugia et al. (2014) carried out a wave-basin test campaign to measure the wind turbine power and wake for a TLP-FOWT subjected to regular waves. Hu et al. (2015) utilized a 1/300 Froude-scaled wind turbine model and a 3-DOF motion simulator in a wind-tunnel to assess the influence of surge motion on structural loads, and its effect on the near wake (x/D<2, PIV measurements). Wind tunnel tests were conducted by Fu et al. (2019) to quantify the effect of pitch and roll oscillations on power output and wake of a turbine scale model. In this case, rotor thrust was not measured. Schliffke et al. (2020) studied the wake (at a distance of 4.6D) of a 2MW FOWT at 1/500 scale with a porous disk model subjected to imposed surge motion. A porous disk model has some inherent limitations: it is not valid to study near wake and does not reproduce the local aerodynamic loads of blades. One goal of the EU H2020 LIFES50+ project was to develop a reliable aerodynamic model of an FOWT rotor to be used in hybrid wave-basin experiments (a numerical rotor is coupled to a physical scale-model of the floating platform, as done by Sauder et al. (2016)). To this purpose, Bayati et al. (2016) investigated the effect of imposed surge and pitch motion on rotor-thrust with a 1/75 scale-model of the DTU 10MW. Measurements were utilized to assess the prediction capabilities of AeroDyn (Moriarty and Hansen (2005)) with respect to FOWTs. The analysis evidenced some differences between simulation and experiment that suggested a further study of the problem. Bayati et al. (2017b) carried out a second wind-tunnel test campaign with focus on the effect of surge motion of the wind turbine wake, that was measured with hot-wire probes.

In parallel with wind-tunnel experiments, floating turbine model tests were performed in different wave-basins. Among the goals of these experiment was to investigate the effect of turbine aerodynamic loads on the global response of the system. Goupee et al. (2012) investigated at 1/50 scale the response of three 5MW FOWTs to wind and wave excitation. The blades of the turbine model were a geometrically scaled version of the NREL 5MW blade, and the aerodynamic performance (thrust and power) of the rotor was not representative of the full-scale turbine. This was found to be a consequence of the Froude-scaled

low-Reynolds wind. To cope with this issue, Goupee et al. (2014) designed a new rotor to carry out a second set of tests. This second campaign proved that wind-turbine aerodynamic loads must be reproduced correctly when assessing the global response of FOWTs in wave-basin tests. More recent research efforts, like the work of Goupee et al. (2017) or of Bredmose et al. (2017), studied the interaction between turbine-control, aerodynamic forces, and platform motion. Overall, integrated wave-basin tests proved to be very useful in studying the coupled response of floating turbines, with simultaneously modeling of wave excitation, wind, and turbine control. However, reproducing the turbine aerodynamic response is hindered by the low-Reynolds number imposed by Froude-scaling (Bayati et al. (2018)) and by quality of the wind environment (Martin et al. (2014)). With these limitations, reproducing a realistic turbine wake is usually out of reach.

The unsteady response of FOWTs is still an open question. In this respect, this article presents the wind-tunnel scale-model experiment that was carried out as part of the IRPWind UNAFLOW project. The goal of the experiment was to study the aerodynamic response and wake for an FOWT subjected to large surge motion, as it normally occurs in operation. Studying these issues at small scale has some limitations because it is not possible to exactly reproduce all the physics of a full-scale system (e.g., structural response, inflow conditions). However, this disadvantage is offset by the possibility to accurately control the test conditions, and to implement more measurements than in a real turbine. The main contributions of this work are as follows:

- a preliminary 2D experiment is performed to characterize the airfoil used in the turbine model blades. Unlike in previous studies, knowledge of the blade airfoil aerodynamic response is leveraged to select the wind and surge-motion conditions for the 3D experiment. In particular, unsteady lift coefficient data were utilized to ensure that angle-of-attack variation due to rigid-body motion do not cause unsteady airfoil aerodynamics. In this way it is possible to say that any turbine unsteady aerodynamic behavior caused by surge motion is due to rotor unsteadiness rather than airfoil-level unsteadiness. In addition, 2D data are a reliable polar-dataset that can be used to create numerical models of the experiment;

- accuracy of force measurements is improved with respect to the previous test campaigns of Bayati et al. (2016) and Bayati et al. (2017b). The flexible tower in LIFES50+ tests created issues in the measurements, making their use difficult for code validation;

- thrust force measurements from full-turbine experiments are compared to predictions of a quasi-steady rotor-disk model. This model is often relied on when building reduced-order FOWT models for control applications (e.g., Lemmer et al. (2020); Fontanella et al. (2020)), and assessing its prediction capabilities is therefore crucial for developing effective controllers. It is found that thrust force response to surge motion follows quasi-steady theory for reduced frequency below 0.5, which corresponds to the frequency range where modes of a typical FOWT are;

- the wind turbine wake is measured with hot-wire probes to describe and quantify the effect of surge-motion on its energy content. PIV measurements are utilized to visualize the blade tip-vortex inside the wake. It is seen that wake energy is increased in correspondence of the motion frequency.

The impact this paper and the UNAFLOW experiment have on research about FOWT unsteady aerodynamics is:

- additional knowledge about the unsteady aerodynamics of an FOWT. In particular, the analysis is carried out with a system engineering vision of the problem, that considers the response of the entire floating system. Its findings may have an impact on blade design, wind turbine control, wake interaction and wind farm control;

- experimental methodology. The UNAFLOW experiment is the result of a joint effort of different research groups, some expert in numerical simulations and some in wind-tunnel experiments. The experiment followed an integrated approach: results of numerical computations and 2D experiments were utilized to design full-turbine experiments, which results were in turn used for validation of numerical tools. Because of these aspects, the experiment can be considered among the most advanced wind-tunnel tests about FOWT unsteady aerodynamics to date;

- database. Differently than the previous test campaigns of Bayati et al. (2016) and Bayati et al. (2017b), the UNAFLOW experiment generated a comprehensive database that covers in a coherent manner blade airfoil polars, rotor-integral forces and near-wake. The database is accessible at:

```
https://doi.org/10.5281/zenodo.4740005
```

The systematic approach of the experiment makes data especially useful for validating numerical tools. Cormier et al. (2018) utilized the UNAFLOW data to assess predictions of a BEM, a free-vortex and a fully-resolved CFD model. A
105 second comparison with numerical tools was recently carried out by Mancini et al. (2020). The UNAFLOW dataset is currently used for the validation of numerical codes in the IEA Wind Task 30 OC6 project.

The structure of the remainder of this paper is as follows. Section 2 describes the approach followed to design the experiment, and to select the wind and surge-motion conditions. Section 3 presents the 2D sectional model tests that were carried out at the Technical University of Denmark (DTU) Red wind tunnel, to characterize the aerodynamic coefficients of the SD7032
airfoil, used in the turbine model blades. Section 4 describes the full-turbine experiment, with emphasis on the wind turbine scale model and measurements that were carried out. Section 5 reports the main findings of the full-turbine experiments with surge-motion, in particular those about rotor-thrust force, the energy content of the near-wake, and the tip-vortex. Section 6 draws the conclusions and gives some recommendations for future research.

## 2   Concept and design of the experiment

FOWTs undergo large rigid-body motions that are due to the high-compliance of the floating foundation and wind/wave excitation. Consequently, an FOWT rotor often operates in unsteady-flow conditions. The UNAFLOW project studied the unsteady behavior of an FOWT rotor. Core of the experiment is an extensive wind tunnel test campaign with a 2.38m-rotor turbine model that was subjected to imposed surge motion. The purpose of the wind tunnel experiment was to provide a large dataset of rotor loads and wake measurements for several wind-turbine operating and motion conditions, selected to
be realistic for a multi-megawatt FOWT. 2D sectional airfoil experiments were carried out prior to the full-turbine test, to characterize the aerodynamic response of the SD7032 airfoil used in the turbine model blades. 2D data were used to guide

selection of the surge-motion amplitude and frequency of full-turbine tests, and constitute a reliable polar data set to support numerical modeling.

## 2.1 Wind conditions

The experiment considered three operating conditions, reported in Tab. 1. No closed-loop control strategy was utilized, rotor speed and collective pitch angle were fixed. At RATED1 and RATED2, the wind turbine is operated at the optimum full-scale value of tip-speed ratio (TSR) and power is extracted with maximum efficiency (i.e., the maximum power coefficient is achieved). Since TSR is the same, the angle of attack (AoA) along the blade is equal in the RATED1 and RATED2 conditions. In the above-rated condition (ABOVE) the TSR is lower and the collective pitch angle is increased, to preserve rated power.

Experiments were carried out in smooth flow, and the turbulence intensity across the test section height was approximately 2%.

**Table 1.** Tested wind turbine operating conditions (V is average wind speed, RS is rotor speed, $\beta$ is blade pitch angle).

| Condition | V [m/s] | RS [rpm] | TSR [-] | $\beta$ [deg] |
|-----------|---------|----------|---------|---------------|
| RATED1    | 2.5     | 150      | 7.5     | 0             |
| RATED2    | 4.0     | 241      | 7.5     | 0             |
| ABOVE     | 6.0     | 265      | 5.5     | 12.5          |

  Figure 1 shows the Reynolds number along the span of the turbine-model blade in the three operating conditions of Tab. 1. The Reynolds number is around 100k for most of the blade span in RATED2 and ABOVE conditions, and 50k in RATED1. 2D airfoil sectional model experiments were carried out to measure the blade airfoil aerodynamic coefficients at Reynolds
numbers representative of the full-turbine experiment.

## 2.2 Motion conditions

  The aim of the experiment was to investigate the unsteady aerodynamics of an FOWT rotor due to surge motion. The unsteady aerodynamic problem is complex and multi-physics: platform motion is driven by wave excitation and depends on the characteristics of the floating platform. To keep the focus on aerodynamics, some simplifying assumptions were made about wave
excitation and the resulting motion. The wind-turbine model was forced to move in surge and the other platform degrees-of-freedom were not considered. Surge motion was selected because it produces an along-wind motion of the structure, which is in turn cause of large variations of the wind speed seen by the rotor. Moreover any point of the rotor moves with the same velocity and the effective wind speed is uniform across the rotor.

  The surge motion $x$ considered in the experiments is mono-harmonic:

$$x(t) = A_s \sin(2\pi f_s t) \,, \tag{1}$$

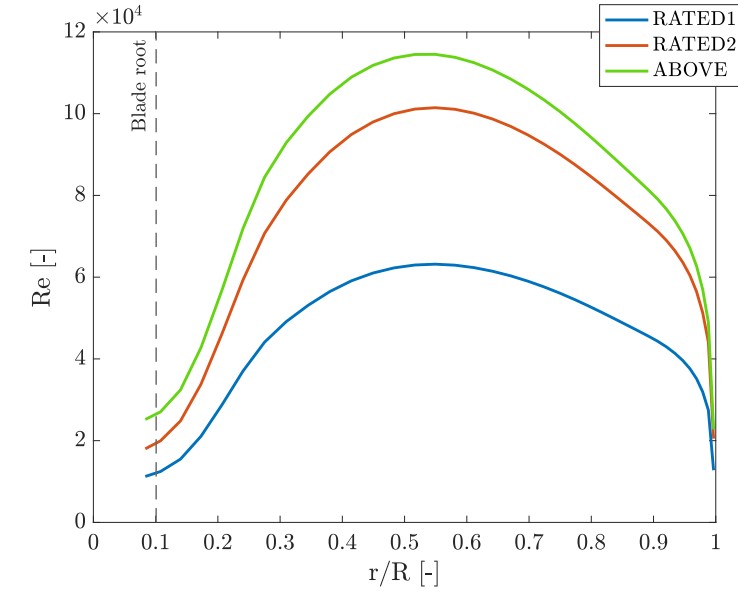

**Figure 1.** Reynolds number as function of non-dimensional radial position (r/R) for the turbine-model blade in the three operating conditions of the experiment. The dashed vertical line indicates the radial position at which the circular cross-section of the blade ends.

where $A_s$ and $f_s$ are the amplitude and frequency of motion respectively. Together with these parameters, surge motion is described by the unit-less reduced-frequency:

$$f_r = \frac{f_s D}{V}, \tag{2}$$

where $V$ is the average wind speed and $D$ the rotor diameter. The experiment investigated several mono-harmonic motion
conditions, obtained from the combination of different values of $A_s$ and $f_s$. Seven frequencies were selected in the range
[0.125 - 2] Hz. The maximum frequency investigated in the full-turbine experiment was limited to 2 Hz to avoid exciting the
model first tower fore-aft (FA) mode. Resonant excitation of this mode may occur due to higher harmonics of the imposed surge
motion, which amplitude decreases with frequency. The frequency of the first FA mode for the turbine model of the previous
LIFES50+ tests (Bayati et al. (2017b)) was 4.25 Hz, and the resonant response penalized force measurements. The UNAFLOW
turbine adopted a stiffer tower with the first FA mode at 6.75 Hz (the scaled frequency of the DTU 10MW is 6.29 Hz) and this
improved measurements accuracy. The selected frequencies are combined with the three wind conditions of Tab. 1 and result
in reduced frequencies between 0.05 and 1.19. Most of previous numerical and experimental research investigated a similar
reduced-frequency range, as documented by Ferreira et al. (2021) (the survey carried out in that paper, currently under review,
defines the reduced-frequency as $k = 2\pi f_s D/V$). A comparison with the surge-motion conditions expected for a full-scale
FOWT is reported in Appendix A.

Four $A_s$ values were tested for each combination of frequency and mean wind speed. Amplitude values were defined based on the maximum surge velocity:

$$\max\left(\dot{x}(t)\right) = \Delta V = 2\pi f_s A_s.$$

(3)

The surge motion causes a variation of the AoA along blades that is, in first approximation, proportional to:

$$\Delta V^* = \frac{\Delta V}{V}.$$

(4)

The four amplitude values were initially selected to achieve, for any pairing of wind speed and motion frequency, $\Delta V^* = 1/20, 3/80, 1/40, 1/80$. At low frequencies, the desired amplitude of motion was beyond the physical limits of the wind tunnel equipment and it was limited consequently. Moreover, amplitude in 1 Hz-frequency cases was increased by 50% to investigate a larger range of $\Delta V^*$.

Figure 2 reports the average AoA along the blade span in the operating conditions of Tab. 1 and the maximum variation caused by the unsteady inflow associated with harmonic surge motion. In steady conditions, a large part of the blade works far from stall, and the variation of AoA due to surge motion is small enough to not overcome this limit. The unsteady aerodynamic behavior of the blade airfoil for harmonic variation of the AoA was characterized with 2D sectional model tests, discussed in the next section. 2D data were leveraged in the design of the full-turbine experiment to assess the blade aerodynamic response caused by surge motion. Figure 2 compares a sample of steady and unsteady lift data with the AoA expected for the turbine blade at 70% rotor radius. AoA variations caused by surge are small enough to keep the blade working in the region where lift is linear. Here, steady and unsteady data are aligned, so no unsteady airfoil response is expected. This ensures any unsteady behavior surge motion may cause in the turbine response is due to rotor unsteadiness rather than airfoil-level unsteady aerodynamics.

## 3   The 2D experiments

2D sectional-model experiments were conducted at the DTU Red wind tunnel to characterize the SD7032 profile behavior in steady and unsteady conditions. Steady experiments, with fixed AoA, provided the airfoil polars for the range of Reynolds numbers explored in full-turbine tests. The experimental polars were used to define the conditions of the full-turbine tests, and also for calibration of numerical models of the experiment (Cormier et al. (2018); Mancini et al. (2020)). Unsteady experiments, with harmonic variation of the AoA, gave an insight into the unsteady aerodynamics of the airfoil. Reproducing the unsteady airfoil behavior with numerical tools was outside the scope of the UNAFLOW project, but a wide dataset of unsteady polars are provided as a project output. This data could be used both for validation of unsteady airfoil aerodynamic models, like it was done by Boorsma and Caboni (2020), or unsteady CFD computations.

The setup for 2D experiments is depicted in Fig. 3. The 2D wing model, of 130mm chord, was fitted at midspan with a pressure loop (32 taps) that was used to measure the lift force from the pressure distribution, and a single-component force transducer that provided an additional lift force gage. The profile drag was obtained by means of a down-stream wake rake.

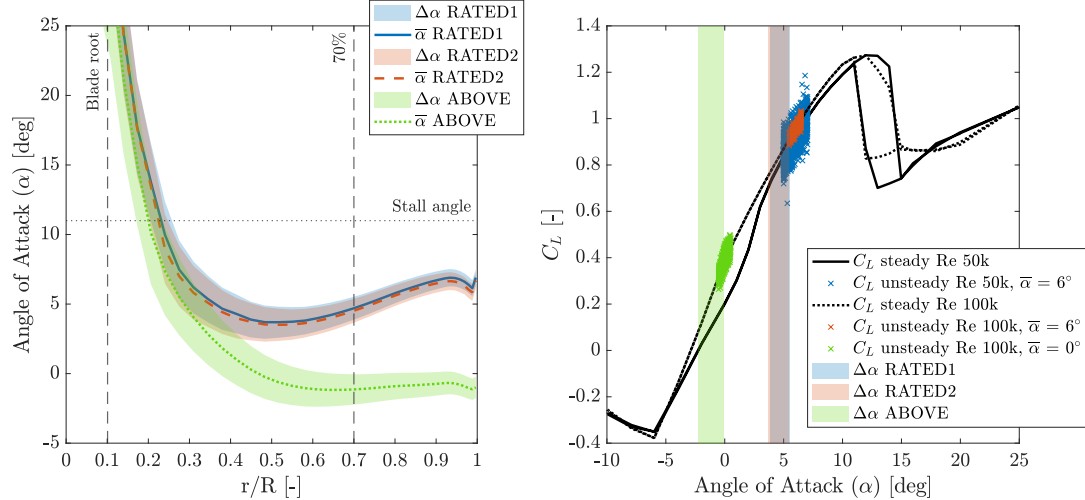

**Figure 2.** Design of the full-turbine experiment based on 2D airfoil data. Left: average angle of attack ($\overline{\alpha}$) and maximum variation due to imposed surge motion ($\Delta\alpha$) as function of non-dimensional radial position (r/R) for the turbine-model blade in the three operating conditions of the experiment. The dashed vertical lines indicate the radial position at which the circular cross-section ends (blade root) and the one at 70% rotor radius). Right: the maximum variation of angle of attack at 70% rotor radius is compared with the steady and unsteady lift coefficient from 2D experiments. Lift data were obtained at Reynolds numbers and angle of attack values representative of those experienced by the turbine model blade in the operating conditions of the full-turbine experiment. Here, unsteady data are aligned with the steady ones and do not show any nonlinear behavior. Hence, surge motion is not expected to cause unsteady airfoil aerodynamics, but rotor-level unsteadiness.

Two ESP 32HD pressure scanners from PSI pressure systems ($\pm$1psi and $\pm$10"H2O range respectively) were connected to the airfoil and wake rake. The profile was mounted on a turning table that set the angle of attack.

### 3.1 Steady force coefficients

Force coefficients were measured for chord Reynolds number of 50k, 60k, 75k, 100k, 150k, 200k and stepping through the AoA range from -10° to 25°. The Reynolds range covers the flow conditions experienced by the turbine model (see Fig. 1). Measurements were repeated in smooth flow (turbulence intensity lower than 0.1%), and with an increased free-stream turbulence that was obtained placing three thin wires (0.15mm diameter) about four chords upstream the profile. The slight increase in turbulence intensity avoids formation of laminar separation bubbles by tripping the boundary layer. The turbulent inflow

condition is deemed to be more realistic and closer to what is experienced by the turbine model blades. The airfoil pressure-lift and wake-drag coefficients are reported in Fig. 4. The lift coefficient shows a non-linear behavior in correspondence of the stall AoA, which is clearly present at Re 50k, and becomes less evident for increasing Re values. The increased turbulence results in a smoother drag coefficient. The effect on the lift coefficient is to smear out the nonlinearity, and this is specially evident for Re values lower than 100k. The wake-drag measurement above 15° is not reliable as the stalled wake covers the entire wake

rake, thus airfoil pressure-based drag is used above stalled AoA. In general the SD7032 lift dependency on Reynolds is low

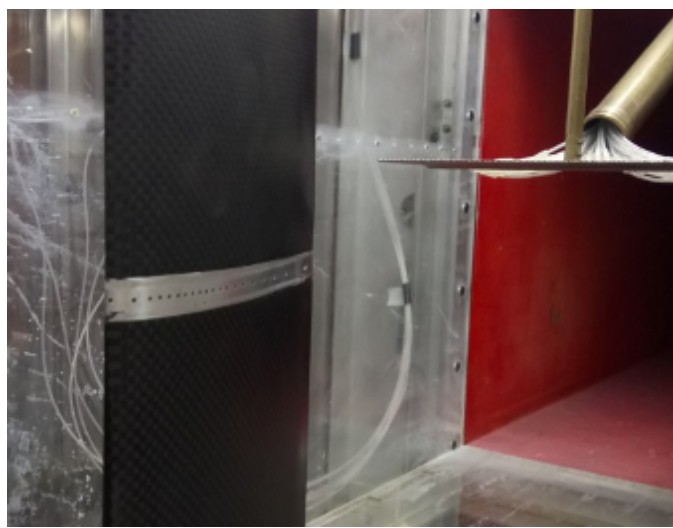

**Figure 3.** The experimental setup used to measure the blade polars in the DTU Red wind tunnel. The pressure loop is at the midspan of the blade sectional model, and the wake-rake is visible downstream.

for AoA 2-11. The linearity of lift is also good and drag does not show any nonlinearity, which are good characteristics for a model-scale rotor.

### 3.2 Unsteady force coefficients

The lift and drag force coefficients were also measured with unsteady airfoil pitching. The conditions of the 2D experiment reflected those of the full-turbine blade. The experiments investigated the profile behavior for chord-Reynolds number of 50k, 100k, 150k, and a static AoA of 0, 3, 6, 9, 10, 12, 15 degrees. The amplitude and frequency of sinusoidal pitching reflects the AoA variation produced by the imposed surge motion in full-turbine tests. The AoA amplitudes were 0.5, 1, 2, 5 degrees and the frequencies 0.25, 0.5, 1, 2, 3 Hz. A sample of results is reported in Fig. 2, with reference to the inflow with increased turbulence, a chord Re of 50k, and a sinusoidal variation of the AoA of $5°$ amplitude and different frequencies. A hysteresis cycle is always present when the airfoil is pitched in correspondence of the stall AoA, and increasing the motion frequency, the strength of this effect is increased. The amplitude of the hysteresis cycle is instead small in the linear region (i.e., for AoA lower than the stall value), where most of the wind turbine model blade operates (see Fig. 2).

### 4 The full-turbine experiments

The full-turbine experiment was carried out at the Politecnico di Milano wind tunnel (Galleria del Vento Politecnico di Milano GVPM). The facility is a closed-loop subsonic wind tunnel and the flow is generated by 14 fans. UNAFLOW tests were carried out in the low-speed test chamber, which has a cross section of 3.84x13.84 m. The wind turbine was developed by Bayati et al. (2017) in the LIFES50+ EU H2020 project as a 1/75 model of the DTU 10MW (Bak et al. (2013)). The turbine rotor of

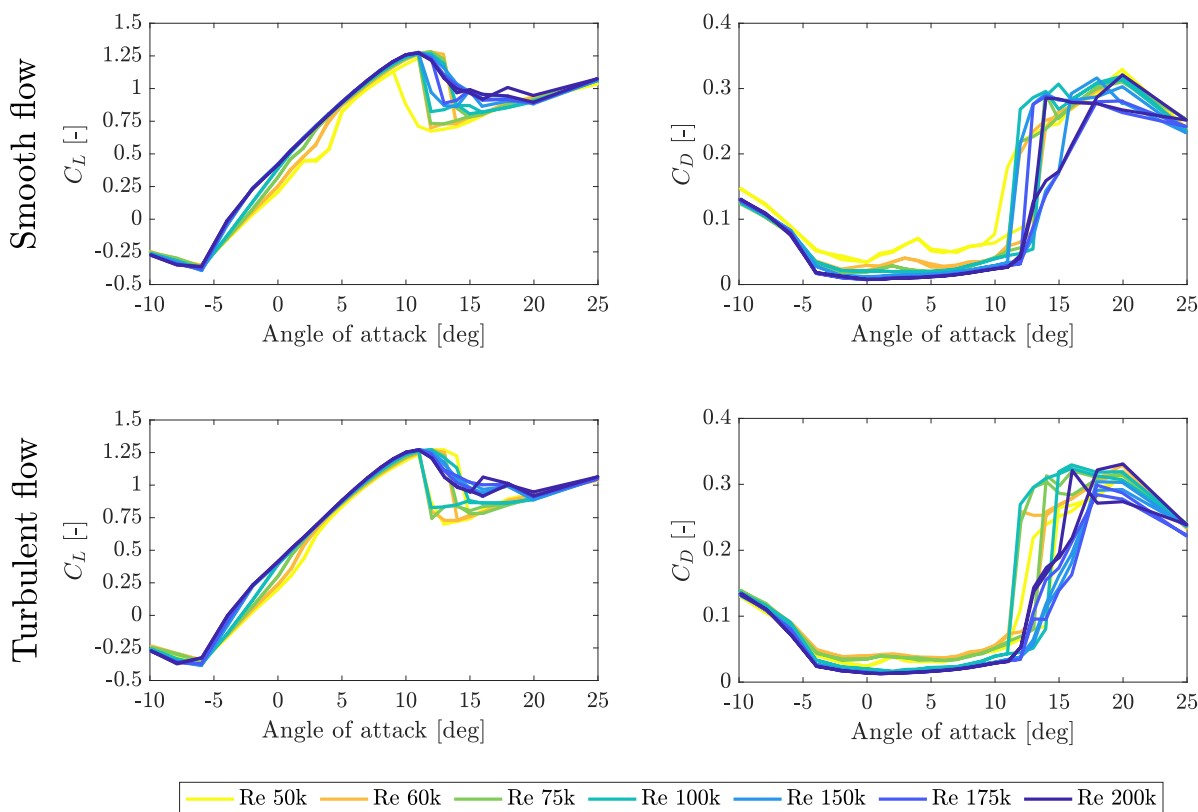

**Figure 4.** Steady lift ($C_L$) and drag ($C_D$) coefficients from 2D sectional model tests in smooth flow (top row) and with added turbulence (bottom row).

2.38m diameter was designed based on performance-scaling (see Kimball et al. (2014)) with the aim of reproducing the thrust and power coefficient of the DTU10MW in 1/3-scale wind. To achieve these goals, the blade was designed using the SD7032

airfoil, modifying the original chord and twist distributions as explained in detail by Bayati et al. (2017a). The scale model specification are reported in Tab. 2.

The turbine scale model was installed on the test rig shown in Fig. 6, which is formed by a slider driven by a hydraulic actuator, and was utilized to simulate surge motion. On top of the slider, there is a second hydraulic actuator connected to the base of the tower through a slider-crank mechanism that was utilized to tilt the wind turbine and make the rotor vertical

(offsetting rotor tilt).

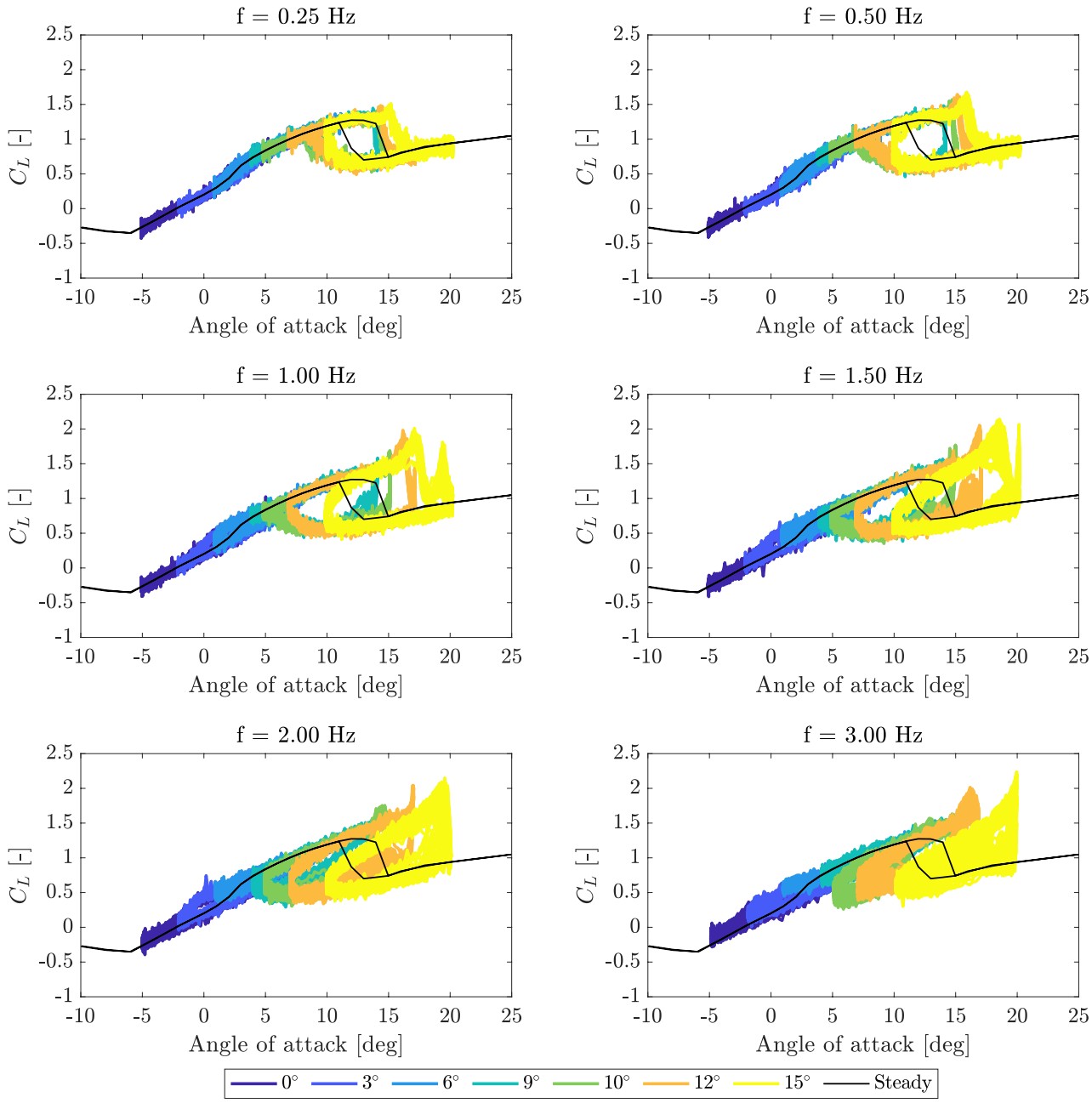

**Figure 5.** Unsteady lift coefficient ($C_L$) for a sinusoidal angle of attack variation of 5° and different frequencies, with turbulent inflow and a chord Re of 50k. Colors denote the mean angle of attack, and the steady lift coefficient is reported in black.

**Table 2.** Specifications of the wind turbine scale model (RNA stands for rotor-nacelle assembly).

| Parameter | Unit | Value |
|---|---|---|
| Rated wind speed | m/s | 3.80 |
| Rated rotor speed | rpm | 240 |
| Rotor diameter | m | 2.38 |
| Blade length | m | 1.10 |
| Hub diameter | m | 0.18 |
| Shaft tilt angle | deg | 5.00 |
| Blade mass | kg | 0.21 |
| Nacelle mass | kg | 1.79 |
| RNA mass | kg | 3.58 |

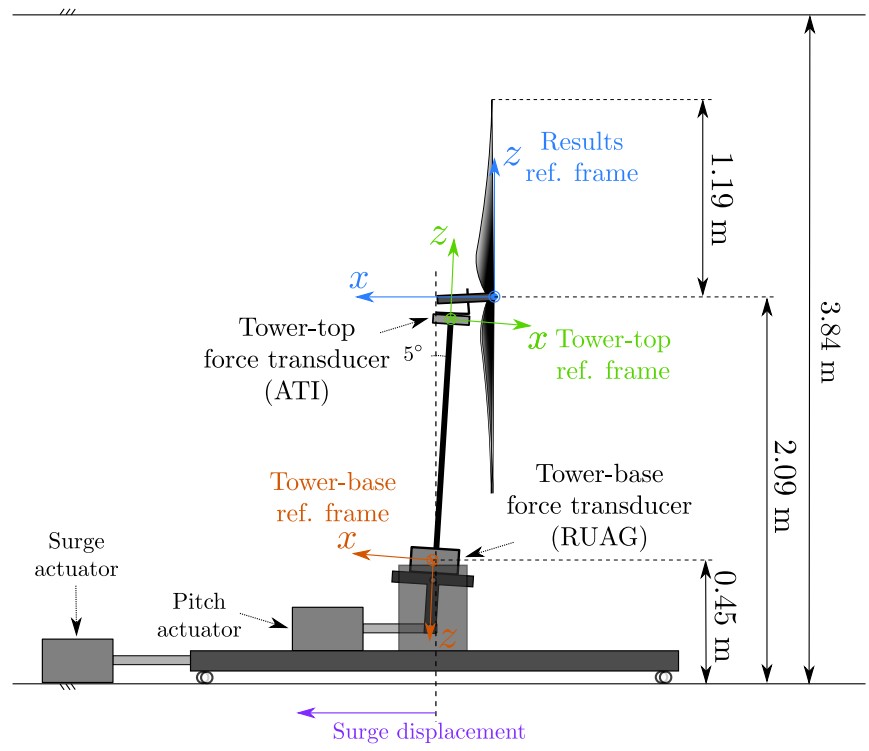

**Figure 6.** Schematic of the full-turbine test setup and coordinate systems used for measurements and their analysis.

## 4.1 Measurements

Several measurements were carried out during the experiment. The undisturbed wind velocity was measured by a pitot tube 7.15m upstream the turbine rotor, centerline, hub-height. An LVDT sensor provided the feedback for the control system of the surge hydraulic actuator. In parallel, the wind turbine surge motion was measured by means of a MEL M5L/200 laser sensor. The tower-top forces were measured by a 6-components force transducer. PCB MEMS accelerometers were fixed in correspondence of the tower-base and nacelle to measure the surge acceleration. All instruments were sampled synchronously with a frequency of 2000 Hz. In few selected test cases, the wake of the wind turbine was scanned by tri-axial hot-wire probes. In an even smaller sample of test cases, PIV measurements were carried out to describe the wake flow structure.

## 4.2 Rotor-integral aerodynamic forces

Rotor-integral aerodynamic forces were evaluated from two load cells, one installed at tower-base (RUAG SG-Balance 192-6i) and one at tower-top (ATI Mini45 SI-145-5). The two sensors and the coordinate systems utilized for force measurements are depicted in Fig. 6. Aerodynamic loads are obtained removing the inertial and weight components from force measurements. Each motion condition of the wind tests (SIW) of Tab. B1-B3 was tested without wind and with fixed rotor (NOW). In the NOW tests, the output of the tower-top sensor is only the inertia and weight of the rotor-nacelle assembly. Aerodynamic forces are then obtained subtracting NOW measurements from SIW measurements:

1. for any given motion condition, the SIW time histories are synchronized with the corresponding NOW;

2. SIW and NOW time histories are trimmed, keeping the maximum number of full periods of motion;

3. the aerodynamic forces are obtained subtracting the NOW time series from the SIW time series:

$$F_{a,i}(t) = F_{\text{SIW},i}(t) - F_{\text{NOW},i}(t) \qquad i = 1, \ldots, 6. \tag{5}$$

The force-subtraction procedure relies on the rigid-body assumption for the tower and blades, hence structural loads depend only on the type of motion and are the same in the NOW and SIW tests. This is valid when the surge-motion frequency is lower than the natural frequencies of the wind turbine components and, in particular, of the first FA mode. For higher motion frequencies, the dynamic amplification associated with tower flexibility cannot be neglected, and results obtained based on the inertia-subtraction procedure may be unreliable. Flexibility of turbine-model components is a source of uncertainty for the experiment, but its quantification was outside the scope of the UNAFLOW test campaign. An additional test campaign is currently planned to address this specific issue.

## 4.3 Hot-wire wake measurements

An automatic traversing system was utilized to measure the three-component velocity in the turbine model wake. The system, depicted in Fig. 7, consists of a moving arm mounting two hot-wire probes. Measurements were carried out with the traversing system spanning across the Y-Z plane (cross-wind, CW) or the X-Z plane (along-wind, AW) of the "Results ref. frame" of

Fig. 6. In the CW case, the measurement plane was 2.3D (5.48m) downwind the turbine. This was the furthest distance allowed by the size of the wind-tunnel test chamber, and it is part of the near-wake region (Vermeer et al. (2003)). One of the probe was mounted at hub-height, the other 0.2m below. The probes were moved in the cross-wind direction, ranging from -1.6m to 1.6m with respect to the hub position, with a distance of 0.1 m between subsequent points. CW measurements were carried out both for the RATED2 and ABOVE conditions, with and without surge motion. In the AW case, probes were mounted at hub-height, one next to the other: the first at y = 0.7m, the second at y = 0.9m. The probes were moved in the along-wind direction, ranging from 2.18m to 5.48m downwind the hub location, with a distance of 0.33m between subsequent points. AW measurements were carried out only for the RATED2 condition, with and without surge motion.

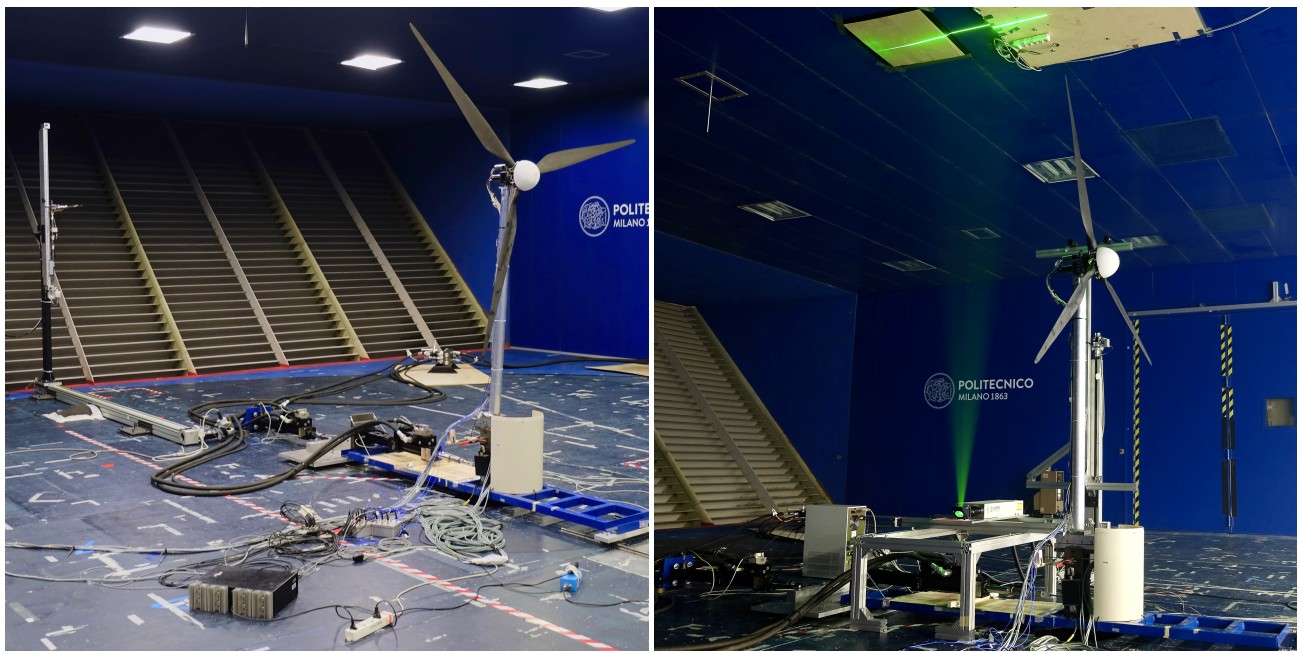

**Figure 7.** Test setup for along-wind (AW) hot-wire measurements (left) and PIV (right) measurements.

### 4.4 PIV wake measurements

A PIV system was used to investigate a portion of the X-Z plane in the near-wake region. The PIV system is made of an Nd:YAG double pulsed laser and two adjacent cameras, mounted on a traversing system, with a line of sight perpendicular to the laser sheet. The measurement area ranged from 0.6m to 1.35m downwind the hub location, and from 0.6m to 1.39m from the hub in the vertical direction. Image pairs were post-processed with PIVTEC PIVview 3C. PIV measurements were carried out for the RATED2 condition, with and without surge motion. For tests without surge motion, measurements were phase-locked to the blade-1 azimuth position ($\psi$). 100 image pairs were acquired for each measurement, from $\psi = 0°$ to $\psi = 120°$ every 15°, and from $\psi = 120°$ to $\psi = 360°$ every 30°. For tests with surge motion, only the motion conditions with a frequency

which is integer sub-multiple of the rotor frequency (i.e., 4 Hz) were considered. Measurements were phase-locked to the surge position, and image pairs were acquired in several points of the motion cycle. Being the rotor frequency an integer multiple of the surge frequency, the blade-1 azimuth position is the same for any measurement in a given surge position.

## 5  Key findings of the full-turbine experiment

This section reports the key findings of the full-turbine experiment. First, rotor-thrust force measurements are compared to the prediction of a quasi-steady model for several harmonic surge motions. Rotor-thrust affects the along-wind response of the floating turbine, and is in turn affected by its motion (i.e., it is a state-dependent force). A correct prediction of thrust-force response to turbine motion is therefore important when assessing the global dynamics of an FOWT. Second, the effects of surge motion on the turbine near-wake are investigated by means of hot-wire measurements. Spectral analysis reveals how surge motion affects the wake energy content. Last, PIV measurements of the wake area near the rotor show effects of turbine translation on blade-tip vortex.

### 5.1  Rotor thrust force

Tower-top force measurements are analyzed to investigate the thrust-force response to surge motion. The analysis is based on a simplified description of the wind turbine rotor, which focusses on integral forces rather than single-blade loads. According to this model, the rotor produces a thrust force:

$$T = \frac{1}{2}\rho \pi R^2 C_T V^2 \,, \tag{6}$$

where $\rho$ is the air density, $R$ the rotor radius, $C_T$ the thrust coefficient and $V$ the undisturbed wind speed. The thrust coefficient is set by the turbine operating condition, which is defined by the TSR $\lambda$ and the collective pitch angle $\beta$:

$$C_T = C_T(\lambda, \beta), \qquad \lambda = \frac{\omega R}{V} \,. \tag{7}$$

The thrust force can be linearized based on a first-order Taylor expansion:

$$T \simeq T_0 + K_{VT}\Delta V + K_{\beta T}\Delta \beta + K_{\omega T}\Delta \omega \,, \tag{8}$$

where $T_0$ is the steady-state thrust force; $\Delta V$, $\Delta \beta$ and $\Delta \omega$ are the variation of wind speed, collective pitch angle and rotor speed from their respective steady-state value; $K_{VT}$, $K_{\beta T}$ and $K_{\omega T}$ are the partial derivatives of thrust with respect to wind speed, collective pitch and rotor speed (the definition is reported for example in the book of Bianchi et al. (2007)). In the present case, collective pitch and rotor speed are fixed, so:

$$T \simeq T_0 + K_{VT}\Delta V \,, \tag{9}$$

with:

$$K_{VT} = \frac{T_0}{V}\left(2 - \left.\frac{\partial C_T}{\partial \lambda}\right|_0 \frac{\lambda_0}{C_{T,0}}\right), \tag{10}$$

where $\lambda_0$ is the steady-state TSR and $C_{T,0}$ the steady-state thrust coefficient. The wind speed seen by any point of the rotor when the turbine undergoes surge motion is:

$$V = V_0 - \dot{x}, \qquad \Delta V = -\dot{x}, \tag{11}$$

where $V_0$ is the mean wind speed. The thrust force is:

$$T \simeq T_0 - K_{VT}\dot{x}, \qquad \Delta T = -K_{VT}\dot{x}. \tag{12}$$

The thrust force variation induced by the surge motion is function of the wind turbine steady-state operational data only. Equation 12 is therefore referred to as quasi-steady theory. According to quasi-steady theory (QST), the thrust force variation depends only on surge velocity.

The focus of force measurements is the surge-frequency aerodynamic-thrust force, extracted from the tower-top force measurements based on the inertia-subtraction procedure presented in Sec. 4.2. The surge-frequency thrust force is:

$$\Delta T = |\Delta T|e^{j\phi}, \tag{13}$$

where $|\Delta T|$ is the amplitude of thrust force at the surge frequency and $\phi$ is the phase with respect to the surge displacement. In general, the surge-frequency thrust force has a component in opposition of phase to surge velocity, and one in opposition of phase to surge acceleration. According to the QST model of Equation 12, thrust force is perfectly aligned to surge velocity. The adherence of thrust force measurements to the QST model is verified computing the unsteady thrust-force coefficient:

$$C_{\Delta T} = \frac{\Delta T}{\frac{1}{2}\rho\pi R^2 V^2}. \tag{14}$$

This non-dimensional quantity is also useful to ease comparison of experimental results to other studies. According to the QST the thrust force variation is:

$$|\Delta T| = 2\pi f_s A_s K_{VT}, \tag{15}$$

and the unsteady thrust coefficient is:

$$C_{\Delta T}^{QST} = 2\pi f_r A_r C_0^*, \tag{16}$$

where $A_r = A_s/D$ is the reduced surge-amplitude, and:

$$C_0^* = \left(C_{T0}\left(2 - \left.\frac{\partial C_T}{\partial \lambda}\right|_0 \frac{\lambda_0}{C_{T0}}\right)\right). \tag{17}$$

The experimental unsteady thrust coefficient for several surge-motion conditions is reported on the left of Fig. 8 as a function of $f_r$ (notice that $C_{\Delta T}$ is divided by $A_r$), and the thrust-force phase is shown on the right of the same figure. Measurements are compared to QST predictions, which correspond in the plot to straight lines, obtained as in Eq. 16. The QST prediction depends on the turbine operating condition and its steady-state thrust coefficient characteristic. According to QST, the phase

is -90°, regardless of the motion condition and wind speed. Measurements where the surge-motion frequency was higher than 1.5 Hz were discarded from the analysis, to exclude any effect of tower flexibility. Uncertainty due to tower flexibility is not quantified, but is deemed small for imposed surge-motion frequencies below 1.5 Hz. For values of reduced frequency below 0.5, thrust force measurements are aligned to QST predictions. For increasing values of reduced frequency, $C_{\Delta T}$ appears to progressively shift away from the QST line, with a trend that is consistent in the RATED2 and ABOVE cases (these results are compared to surge motion conditions experienced by a full-scale turbine in Appendix A). The same trend cannot be easily identified in the phase $\phi$, as data are scattered in a range of $\pm 10°$ around -90°. The uncertainty in phase data is related to the thrust force component opposed to surge acceleration which is responsible of phase deviations and it is difficultly measured. Analysis of uncertainty related to flexibility of turbine model components may help discerning if the deviation of data from QST for increased surge-motion frequency is due to tower flexible dynamics or rotor aerodynamic response.

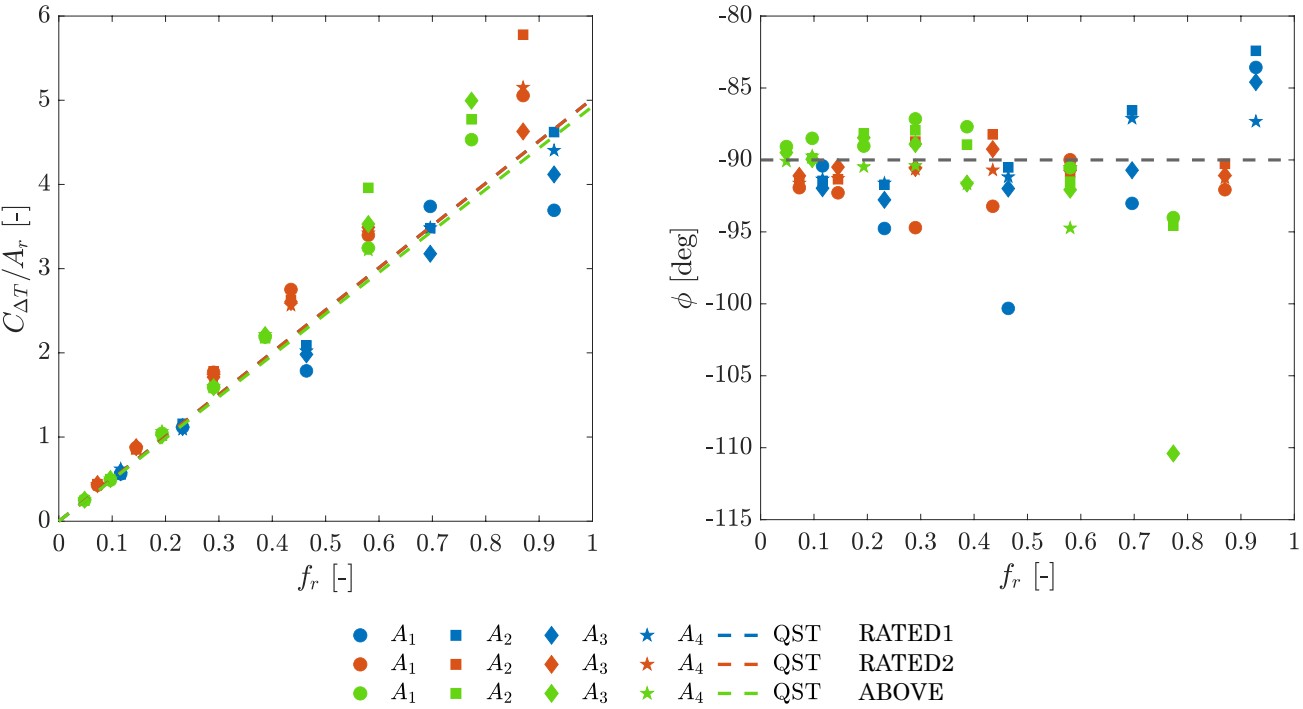

**Figure 8.** Unsteady thrust coefficient (left) and phase of the thrust-force with respect to surge motion (right) against reduced frequency. Color is for mean wind speed, marker for surge amplitude (e.g., for RATED1 and $f_r = 0.928$, $\bullet = 0.008$m, $\blacklozenge = 0.015$m, $\blacksquare = 0.025$m, and $\bigstar = 0.030$m, see Tab. B1-B3), dashed line for quasi-steady theory prediction.

## 5.2 Hot-wire wake measurements

The wake shape at hub-height is captured by the mean velocity deficit. The deficit for all the conditions that were investigated with hot-wire measurements is shown in Fig. 9. The reduction of axial velocity is always higher for RATED2, where the wind

turbine is operated at maximum power coefficient, compared to ABOVE. The wake is also slightly asymmetric with respect to the hub. For any condition, the velocity deficit is larger on the left side compared to the right. A speed-up is observable at the wake extremities, which is caused by wind tunnel blockage. With surge motion, the wake is slightly narrower, meaning there is more energy in its outer region. A part from that, surge does not change significantly the shape of wake deficit. Even if there are some major differences in the experiment (a porous disk was used to emulate the wind turbine rotor, measurements were carried out at a distance of 4.6D, the inflow was turbulent), this is in agreement with what is found by Schliffke et al. (2020).

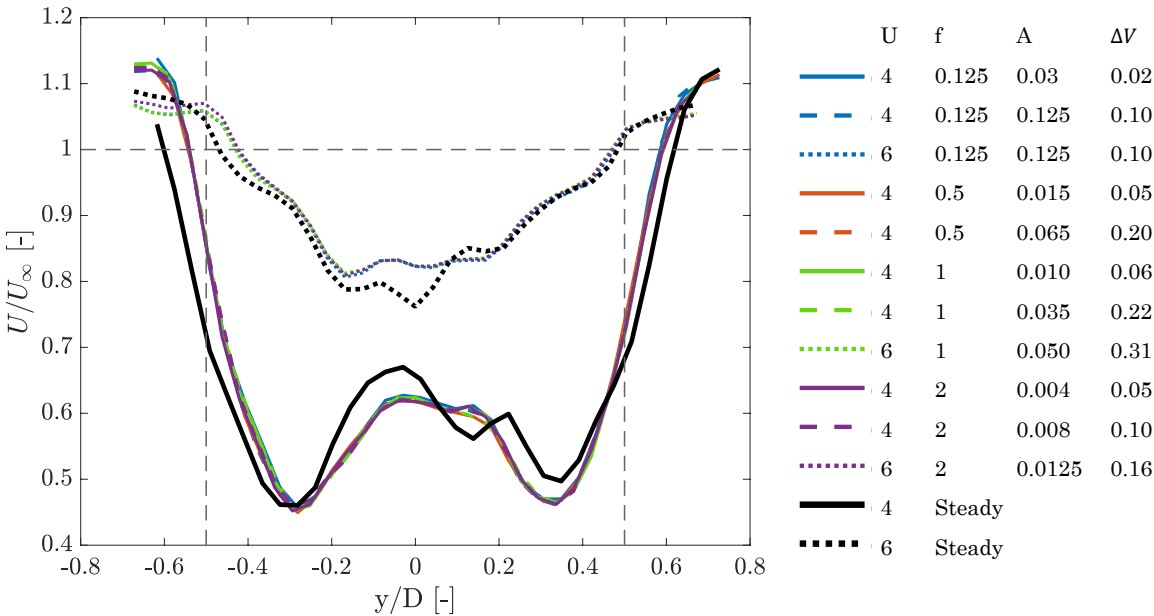

**Figure 9.** Mean velocity deficit at hub-height ($y/D = 0$ corresponds to the hub location, the rotor edge is marked by the vertical dashed lines) for the steady cases (i.e., without surge motion) and unsteady cases. In the legend: U is the mean wind speed, f the surge frequency, A the surge amplitude, $\Delta V$ the maximum surge velocity.

The wake frequency content at hub-height is studied computing the PSD of the three velocity components across the wake. Figure 10 compares the spectra for the RATED2 case without and with surge motion, in particular with reference to the case of $f_s = 1$ Hz, $A_s = 0.035$ m. The energy content is concentrated in the outer region of the rotor and it is reasonably related to the blade-tip vortex. This distribution of energy is common also to any RATED2 case. The asymmetry seen in the velocity
deficit is found also in spectra, and it is particularly evident in the vertical component, which is associated with rotor spinning. Looking at the unsteady case, a strong harmonic component is visible at the frequency of motion, which is absent in the steady case. The harmonic at surge-frequency is more evident in the axial velocity, compared to the other two velocity components. Another strong harmonic component is visible close to $f = 4$ Hz, the 1P frequency, and it is associated with aerodynamic imbalance (i.e., slightly different pitch settings for the three blades).

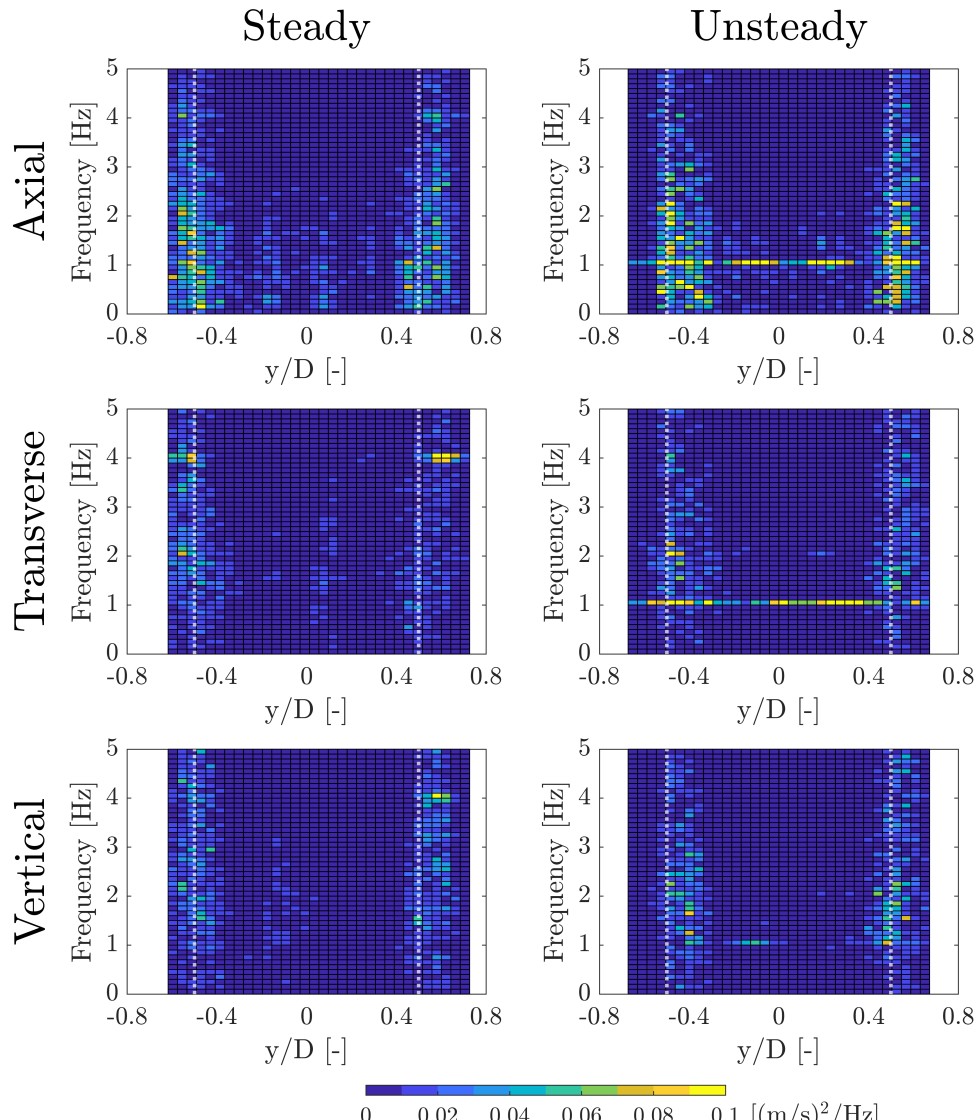

**Figure 10.** PSD of the hub-height wake velocity components in several cross-wind positions ($y/D = 0$ corresponds to the hub location, rotor edge is marked by the vertical dotted lines) for the RATED2 case. The surge motion of the unsteady case is with $f_s = 1$ Hz, $A_s = 0.035$ m.

The same analysis is carried out in Fig. 11 for one ABOVE condition. In this case, energy is concentrated in the inner region of the rotor, witnessing the presence of a strong blade-root vortex. Also in this case, the 1P component is visible, at $f = 4.417$ Hz and the wake is slightly asymmetric. In case of surge motion, the harmonic at surge frequency becomes dominant.

More information about surge motion effects on wake is provided by two additional metrics obtained from the axial velocity spectrum. The space-averaged PSD is a description of energy distribution in frequency and it is obtained summing summing

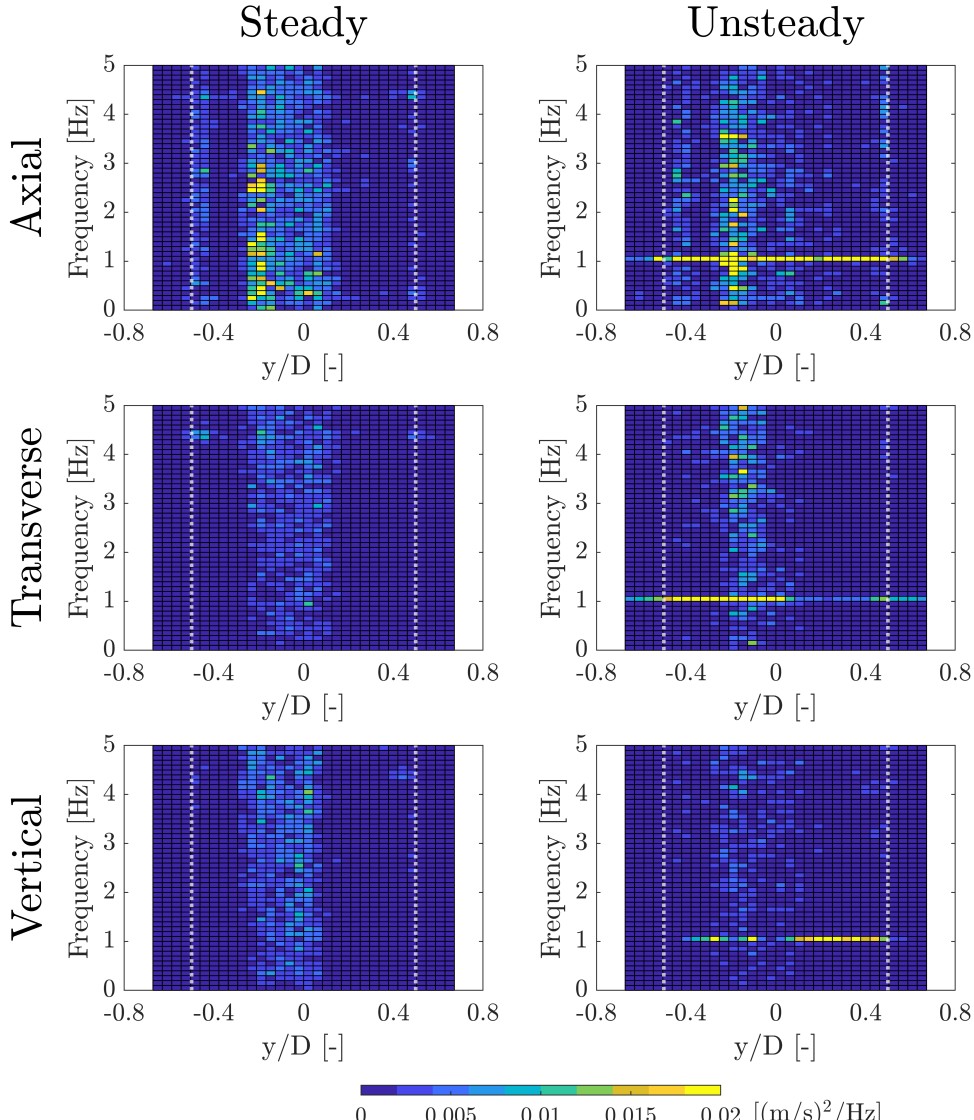

**Figure 11.** PSD of the hub-height wake velocity components in several cross-wind positions ($y/D = 0$ corresponds to the hub location, rotor edge is marked by the vertical dotted lines) for the ABOVE case. The surge motion of the unsteady case is with $f_s = 1$ Hz, $A_s = 0.05$ m.

across the x-axis of Fig. 10-11:

$$\overline{U}_f = \frac{\sum_{y=1}^{n_y} U_{y,f}}{\sum_{y=1}^{n_y} \sum_{f=1}^{n_f} U_{y,f}^0}, \tag{18}$$

where $U_{y,f}$ is the PSD of the axial velocity at point $y$ evaluated at frequency $f$, $n_y$ is the number of points where the wake speed is measured, and $n_f$ the number of discrete frequencies where the PSD is computed. $U_{y,f}^0$ denotes the PSD for the steady case with the same mean wind speed of $U_{y,f}$. The space-averaged PSD $\overline{U}_f$ for the investigated conditions is shown in Fig. 12.

In the steady case, energy is evenly spread below 1 Hz, and decreases smoothly increasing frequency. A peak is always present at the 1P frequency. The energy is greater in RATED2 compared to ABOVE. The spectrum for any unsteady case is similar to the corresponding steady case, except for a peak at the surge frequency. This suggests some energy is transferred in the wake by the turbine motion. Similar findings, but for the far-wake of a porous disk, are reported by Schliffke et al. (2020). Looking at the PSD of Fig. 12 it is also interesting to notice that, for a surge frequency up to 1 Hz, the amplitude of the surge-frequency

peak is proportional to $\Delta V$, but not linearly. The energy increment in the 2 Hz case is much lower than for any other motion condition with similar $\Delta V$. The surge motion also amplifies the 1P harmonic and the amplification in RATED2 is greater than in ABOVE conditions.

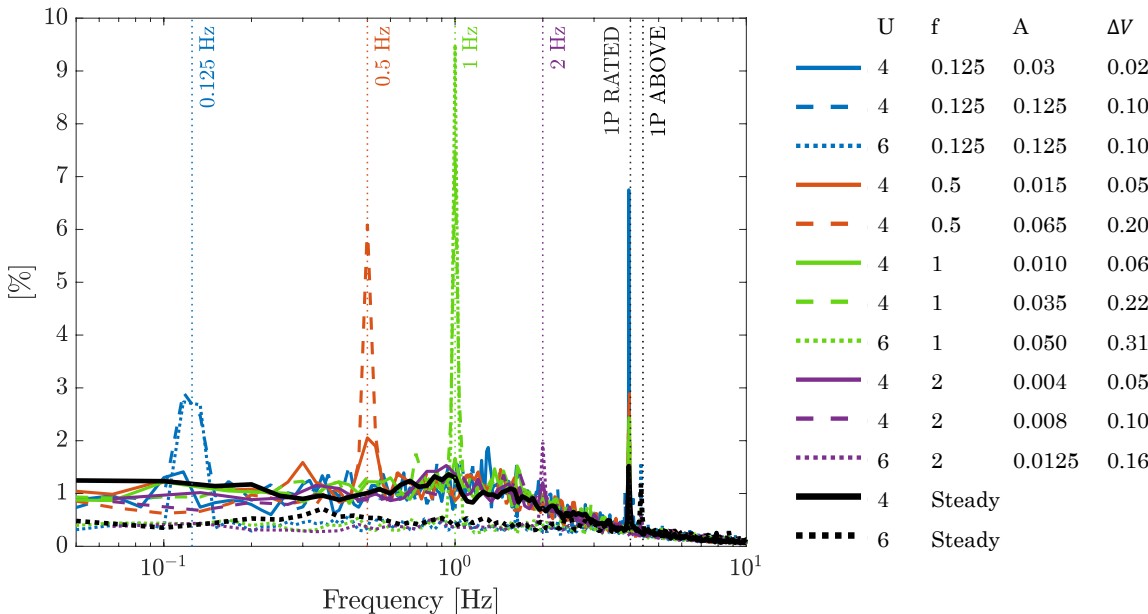

**Figure 12.** Space-averaged PSD of the hub-height axial velocity for different frequencies. The vertical dotted lines mark the frequencies of surge motion and the rotor frequency. In the legend: U is the mean wind speed, f the surge frequency, A the surge amplitude, $\Delta V$ the maximum surge velocity.

The frequency-averaged PSD describes how energy is distributed across the wake and it is computed, for any measurement point, as the frequency-integral of the corresponding PSD (i.e., summing across the y-axis of Fig. 10-11):

$$380 \quad \overline{U}_y = \frac{\sum_{f=1}^{n_f} U_{y,f}}{\sum_{y=1}^{n_y} \sum_{f=1}^{n_f} U_{y,f}}. \tag{19}$$

In this case, $U_{y,f}$ is used for normalization. The frequency-averaged PSDs $\overline{U}_y$ are reported in Fig. 13. The energy space distribution is not affected by surge motion, and it is strictly characteristic of the operating condition. In RATED2, energy is concentrated in the outer region of the rotor and it is associated with blade-tip vortex. In ABOVE conditions, most of the energy

is in the central part of the rotor, where the blade-root vortex is, whereas the contribution of the tip vortex is lower. More energy
is present on the left than on the right of the hub and this is particularly evident in ABOVE cases. Energy is increased across the entire wake, but the increment is more consistent in correspondence of the blade-tip for RATED cases, and the blade-root for ABOVE cases. This suggest that surge motion increases the axial travel velocity of the blade-tip and blade-root vortices.

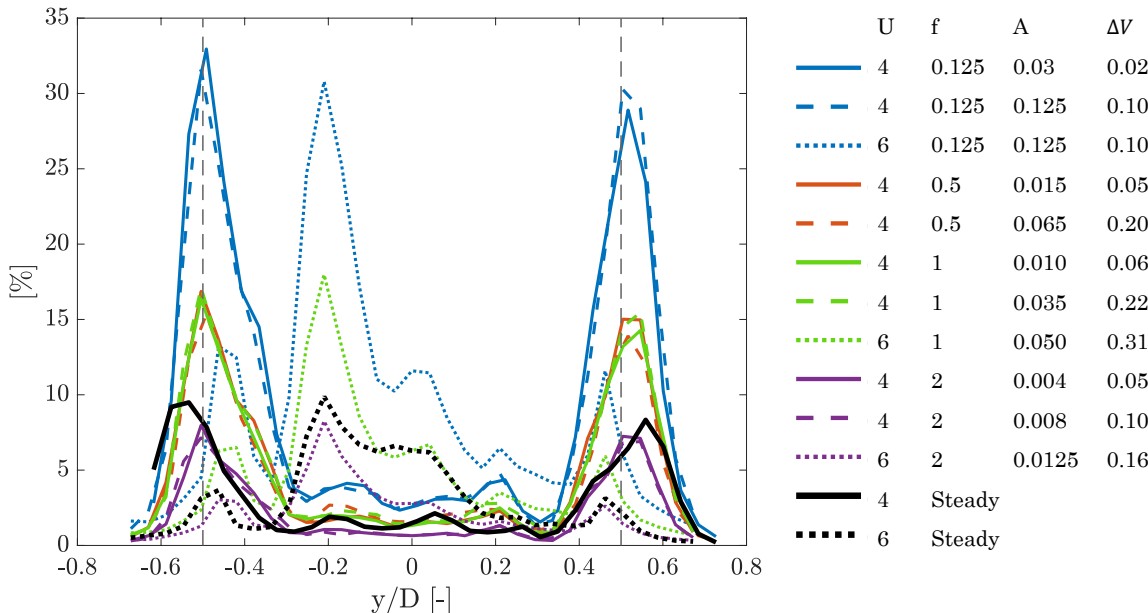

**Figure 13.** Frequency-averaged PSD of the hub-height axial velocity in different cross-wind positions ($y/D = 0$ corresponds to the hub location, the rotor edge is marked by the vertical dashed lines). In the legend: U is the mean wind speed, f the surge frequency, A the surge amplitude, $\Delta V$ the maximum surge velocity.

## 5.3 PIV wake measurements

PIV combined with a realistic turbine model allows to appreciate the wake flow-structures and to investigate how these are
390 affected by turbine motion. The focus of the analysis is on the blade-tip vortex, because it holds a significant fraction of the wake energy, as seen from hot-wire measurements. The blade-tip vortex is visualized from the vorticity magnitude, computed based on the transverse and vertical velocities. Figure 14 reports the vorticity magnitude for the area of the wake near blade-tip without surge motion and with blade-1 at zero degrees azimuth (i.e, vertical and pointing upwards). The tip-vortices shed by the blades are clearly seen. The vortices position does not change for subsequent PIV images captured in the same azimuthal
position of blade-1.

Figure 15 shows the vorticity magnitude in the same condition, but with a surge-motion frequency of 1 Hz and amplitude of 0.065 m. PIV images are acquired in eight different surge positions for a blade-1 azimuth of zero degrees. The tip-vortices

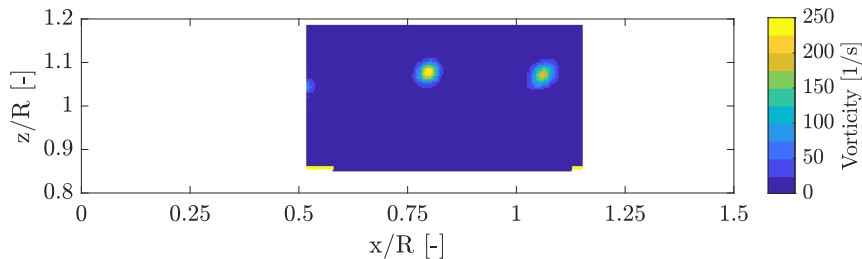

**Figure 14.** Vorticity for the RATED2 operating condition without surge motion. x/R and z/R are the axial and vertical distance from rotor apex normalized by rotor radius. The origin of the axes is coincident with the rotor apex when the wind turbine is in the zero-surge position (see the Results ref. frame of Fig. 6).

position is modified by turbine motion, and varies periodically with its frequency. The mechanism behind wake evolution is explained comparing two phases of the surge cycle with equal but opposite velocity (e.g., phase 4 and phase 8 in Fig. 15). When the rotor moves with downwind velocity (phase 4), the tip vortices are released with a lower velocity than without surge motion and travel a lower distance in the wake. The opposite is true when the rotor moves with upwind velocity (phase 8). An algorithm for detection of vortex position and size, like those presented by Chakraborty et al. (2005), may be used in future to quantify the effect of surge motion on the blade-tip vortex travel speed. The behavior of the tip vortex was studied by means of CFD simulations by Cormier et al. (2018) with similar findings. Numerical simulations show a stable vortex merging which is not evidenced by the experiment.

## 6 Conclusions

This article presented an extensive wind-tunnel experiment for the unsteady aerodynamic response of a floating wind turbine subjected to surge motion. The low-Reynolds airfoil of the turbine-model blade was characterized in a dedicated 2D experiment, in steady and unsteady conditions. The steady lift force coefficient has a linear behavior for AoA between -5 and +8 degrees. A hysteresis cycle is present in correspondence of the stall AoA, when the airfoil is subjected to sinusoidal pitching, and extends to lower AoAs for increasing pitching frequency. Knowledge about the airfoil response is leveraged to select the wind and motion conditions of the full-turbine experiment. Three wind speeds are selected: two are representative of below-rated operations, where the blade is operated at high AoA, one of above-rated, where angle of attack is lower. The turbine model is subjected to harmonic surge motion of several amplitudes and frequencies, selected to produce AoA variations confined in the linear lift-coefficient region, and avoid unsteady airfoil response. Thrust force measurements are carried out to study the full-turbine aerodynamic response to surge motion. Experimental data are compared to predictions of a quasi-steady model to assess the presence of unsteady effects. It is found that data are aligned to quasi-steady theory predictions up to a reduced frequency of 0.5. Above this frequency, unsteady effects may be present. However, a thorough assessment of the experimental uncertainty, in particular the fraction related to flexible response of the turbine tower, needs to be carried out to confirm the unsteady

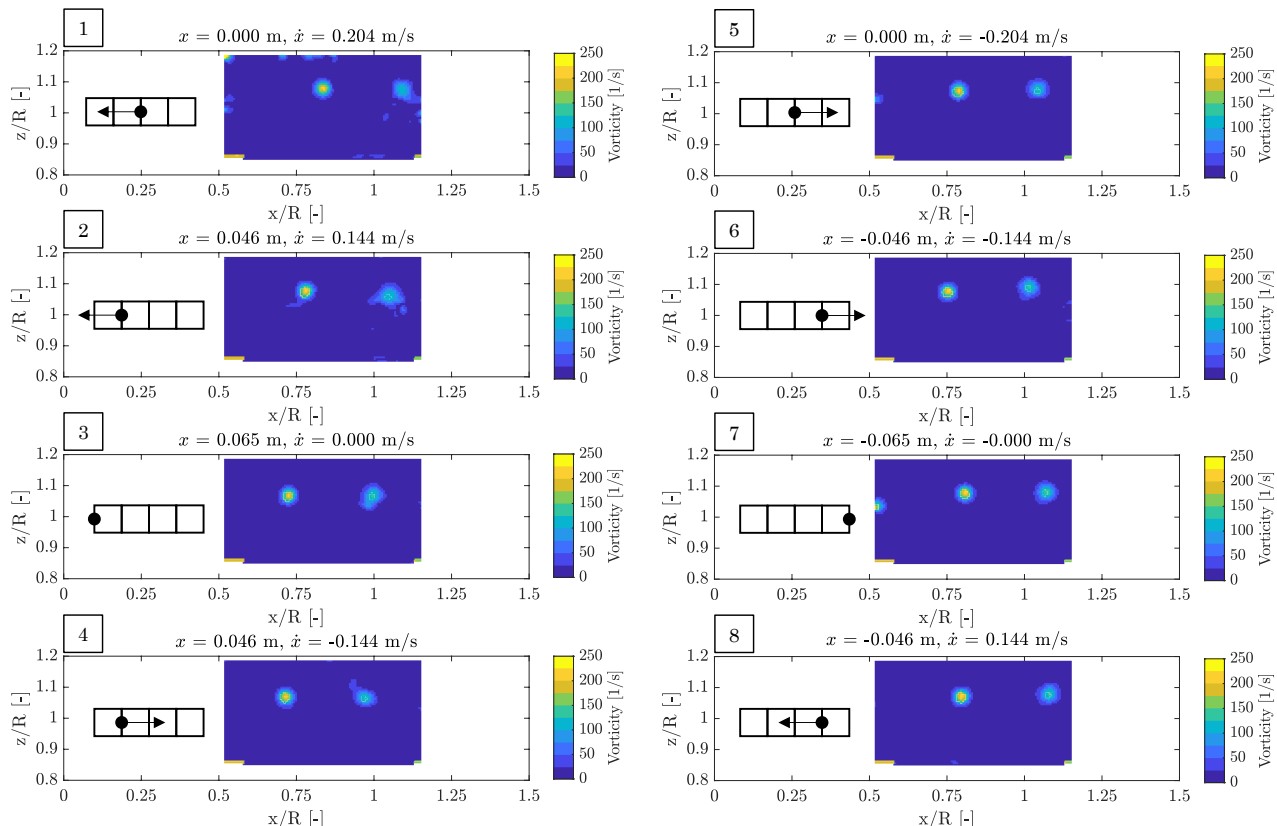

**Figure 15.** Vorticity for eight subsequent wind turbine positions in a surge cycle of frequency 0.5 Hz, amplitude 0.065 m, and RATED2 conditions. The blade-1 azimuth is always zero degrees. x/R and z/R are the axial and vertical distance from rotor rotor apex normalized by rotor radius. The origin of the axes is coincident with the rotor apex when the wind turbine is in the zero-surge position (see the Results ref. frame of Fig. 6).

aerodynamic response for higher surge-motion frequencies. Near-wake measurements were performed with hot-wire probes to assess the effect of surge motion on the wind turbine wake. The average hub-height velocity deficit with surge movement is the same as for the bottom-fixed case. The wake spectral content is increased in correspondence of the surge-motion frequency: the increment (up to 9% compared to the steady case) is proportional to the maximum surge velocity. A spatial analysis suggests that the largest increment is in the outer region of the rotor in RATED conditions, and in correspondence of the most loaded

sections of the blade in the ABOVE condition. PIV measurements phase-locked to the turbine position in the surge cycle and to the rotor azimuth, show that surge motion modifies the travel speed of the blade-tip vortex, that varies periodically with the surge-motion frequency.

The experiment posed some research questions that are still open and could be answered with further investigation:

- the platform pitch response is tightly coupled with rotor aerodynamic response. Because of this coupling, closed-loop pitch-to-feather control strategies may lead to an unstable response of the system (see for example the work of Larsen and Hanson (2007); Jonkman (2008); van der Veen et al. (2012)). A correct description of this coupling is essential to improve the current control methodologies. Moreover, numerical simulations like those of Wise and Bachynski (2020) have shown that pitch motion has a strong influence on vertical wake deflection, and this phenomenon has the potential to be exploited for farm control purposes (Nanos et al. (2020)). Surge and pitch motion are similar as both cause along-wind motion of the rotor, but occur at different frequencies. Moreover, in the surge case, the variation of wind speed across the rotor is uniform, whereas in the pitch case, the flow is skewed. Future wind-tunnel experiments should focus on platform pitch motion;

- unsteady aerodynamic effects appear to be more relevant for increased reduced-frequency of surge motion. It would be worth investigating surge-motion frequencies higher than those considered in this experiment. This can be complicated by the flexible response of the turbine model. The turbine fore-aft mode is set by tower stiffness and weight of the rotor-nacelle assembly. Tower frequency is increased by reducing the latter and increasing the former. Slight stiffness increments are possible modifying the tower design, whereas RNA mass is heavily constrained by mass of actuators (generator and pitch), control electronics and sensors, that are commercial components and cannot be modified. Numerical models can support experiments and circumvent the limitations of the latter. Numerical tools may be validated based on the experimental data already available, and then used to study those conditions that may be unpractical to explore with experiments;

- quantifying uncertainty of the experiment results is important to interpret them correctly. The UNAFLOW dataset is currently utilized in the OC6 project, and one of the project goals is to quantify uncertainty of data used for codes validation. A test campaign like the one discussed in this paper but dedicated to uncertainty quantification is currently planned. Uncertainty could be assessed with a methodology similar to that used by Robertson et al. (2020) for wave-basin tests.

*Data availability.* The dataset of the UNAFLOW experiment is accessible at `https://doi.org/10.5281/zenodo.4740005`.

## Appendix A: Extension of thrust response results to a 10MW floating turbine

Experimental results of section 5.1 indicate the thrust force response follows quasi-steady theory for surge-motion reduced frequency up to 0.5. Here, results are extended to a generic 10MW floating turbine (rotor diameter $D = 178.4$m), for which reduced frequency as function of surge-motion frequency and wind speed is reported in Fig. A1. For semisubmersible and spar platforms, the surge response is in large part at frequencies lower than 0.3 Hz. In the 0-0.05 Hz frequency range, indicated as low, surge motion is dominated by resonant response of the associated mode (e.g., 0.009Hz for the LIFES50+ Nautilus 10-MW semisubmersible and 0.010Hz for the NTNU 10-MW spar considered by Wise and Bachynski (2020)). In the 0.05-0.2 Hz range surge response is driven by wave excitation that is large as most of wave energy is in this frequency range. In the low frequency range, reduced frequency is up to 1, and lower than 0.5 for above-rated winds (>11.4 m/s) (dotted region of Fig. A1). Wind-tunnel results indicate that surge causes a minimal unsteady aerodynamic behavior at these frequencies. Unsteadiness may be instead present in the upper wave frequency range, in particular with moderate wind speed.

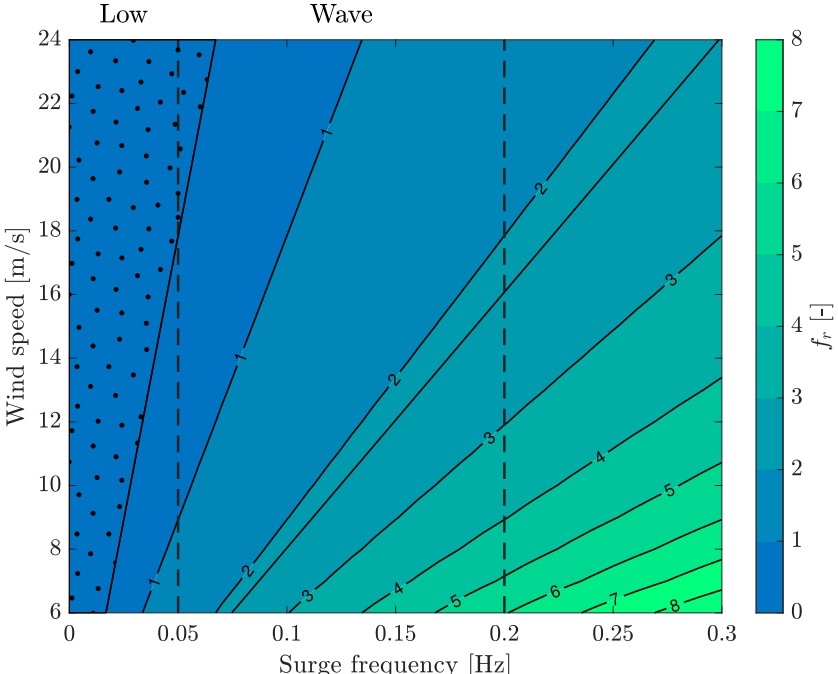

**Figure A1.** Reduced frequency $f_r$ as function of surge-motion frequency and wind speed for a 10MW floating turbine (rotor diameter 178m). The vertical dashed lines mark the low frequency range (0-0.05 Hz) and the wave frequency range (0.05-0.2 Hz). Reduced frequency is lower than 0.5 in the dotted region.

## Appendix B: Test matrices

**Table B1.** RATED1 test matrix. $V$ is the mean wind speed, $f_s$ the surge frequency, $A_s$ the surge amplitude, $f_r$ the reduced frequency, TSR is tip-speed ratio, RS rotor speed, $\beta$ is the blade pitch angle. The CW, AW, PIV columns indicates wether if a cross-wind, along-wind or PIV measurement of the wake was carried out.

| $V$ [m/s] | $f_s$ [Hz] | $A_s$ [m] | $f_r$ [-] | TSR [-] | RS [rpm] | $\beta$ [deg] | CW | AW | PIV |
|---|---|---|---|---|---|---|---|---|---|
| 2.5 | 0.125 | 0.125 | 0.119 | 7.5 | 150 | 0 | | | |
| 2.5 | 0.125 | 0.120 | 0.119 | 7.5 | 150 | 0 | | | |
| 2.5 | 0.125 | 0.080 | 0.119 | 7.5 | 150 | 0 | | | |
| 2.5 | 0.125 | 0.040 | 0.119 | 7.5 | 150 | 0 | | | |
| 2.5 | 0.250 | 0.080 | 0.238 | 7.5 | 150 | 0 | | | |
| 2.5 | 0.250 | 0.060 | 0.238 | 7.5 | 150 | 0 | | | |
| 2.5 | 0.250 | 0.040 | 0.238 | 7.5 | 150 | 0 | | | |
| 2.5 | 0.250 | 0.020 | 0.238 | 7.5 | 150 | 0 | | | |
| 2.5 | 0.500 | 0.040 | 0.476 | 7.5 | 150 | 0 | | | |
| 2.5 | 0.500 | 0.030 | 0.476 | 7.5 | 150 | 0 | | | |
| 2.5 | 0.500 | 0.020 | 0.476 | 7.5 | 150 | 0 | | | |
| 2.5 | 0.500 | 0.010 | 0.476 | 7.5 | 150 | 0 | | | |
| 2.5 | 0.750 | 0.030 | 0.714 | 7.5 | 150 | 0 | | | |
| 2.5 | 0.750 | 0.020 | 0.714 | 7.5 | 150 | 0 | | | |
| 2.5 | 0.750 | 0.015 | 0.714 | 7.5 | 150 | 0 | | | |
| 2.5 | 0.750 | 0.007 | 0.714 | 7.5 | 150 | 0 | | | |
| 2.5 | 1.000 | 0.030 | 0.952 | 7.5 | 150 | 0 | | | |
| 2.5 | 1.000 | 0.025 | 0.952 | 7.5 | 150 | 0 | | | |
| 2.5 | 1.000 | 0.015 | 0.952 | 7.5 | 150 | 0 | | | |
| 2.5 | 1.000 | 0.008 | 0.952 | 7.5 | 150 | 0 | | | |
| 2.5 | 1.500 | 0.015 | 1.428 | 7.5 | 150 | 0 | | | |
| 2.5 | 1.500 | 0.010 | 1.428 | 7.5 | 150 | 0 | | | |
| 2.5 | 1.500 | 0.007 | 1.428 | 7.5 | 150 | 0 | | | |
| 2.5 | 1.500 | 0.0035 | 1.428 | 7.5 | 150 | 0 | | | |
| 2.5 | 2.000 | 0.010 | 1.904 | 7.5 | 150 | 0 | | | |
| 2.5 | 2.000 | 0.007 | 1.904 | 7.5 | 150 | 0 | | | |
| 2.5 | 2.000 | 0.005 | 1.904 | 7.5 | 150 | 0 | | | |
| 2.5 | 2.000 | 0.0025 | 1.904 | 7.5 | 150 | 0 | | | |

**Table B2.** RATED2 test matrix. $V$ is the mean wind speed, $f_s$ the surge frequency, $A_s$ the surge amplitude, $f_r$ the reduced frequency, TSR is tip-speed ratio, RS rotor speed, $\beta$ is the blade pitch angle. The CW, AW, PIV columns indicates wether if a cross-wind, along-wind or PIV measurement of the wake was carried out.

| $V$ [m/s] | $f_s$ [Hz] | $A_s$ [m] | $f_r$ [-] | TSR [-] | RS [rpm] | $\beta$ [deg] | CW | AW | PIV |
|---|---|---|---|---|---|---|---|---|---|
| 4.0 | 0.125 | 0.125 | 0.074 | 7.5 | 241 | 0 | × | × | × |
| 4.0 | 0.125 | 0.100 | 0.074 | 7.5 | 241 | 0 | | | |
| 4.0 | 0.125 | 0.065 | 0.074 | 7.5 | 241 | 0 | | | |
| 4.0 | 0.125 | 0.030 | 0.074 | 7.5 | 241 | 0 | × | | × |
| 4.0 | 0.250 | 0.125 | 0.149 | 7.5 | 241 | 0 | | | |
| 4.0 | 0.250 | 0.100 | 0.149 | 7.5 | 241 | 0 | | | |
| 4.0 | 0.250 | 0.065 | 0.149 | 7.5 | 241 | 0 | | | |
| 4.0 | 0.250 | 0.030 | 0.149 | 7.5 | 241 | 0 | | | |
| 4.0 | 0.500 | 0.065 | 0.298 | 7.5 | 241 | 0 | × | × | × |
| 4.0 | 0.500 | 0.050 | 0.298 | 7.5 | 241 | 0 | | | |
| 4.0 | 0.500 | 0.035 | 0.298 | 7.5 | 241 | 0 | | | |
| 4.0 | 0.500 | 0.015 | 0.298 | 7.5 | 241 | 0 | × | | × |
| 4.0 | 0.750 | 0.040 | 0.446 | 7.5 | 241 | 0 | | | |
| 4.0 | 0.750 | 0.030 | 0.446 | 7.5 | 241 | 0 | | | |
| 4.0 | 0.750 | 0.020 | 0.446 | 7.5 | 241 | 0 | | | |
| 4.0 | 0.750 | 0.010 | 0.446 | 7.5 | 241 | 0 | | | |
| 4.0 | 1.000 | 0.050 | 0.595 | 7.5 | 241 | 0 | | | |
| 4.0 | 1.000 | 0.035 | 0.595 | 7.5 | 241 | 0 | × | × | × |
| 4.0 | 1.000 | 0.025 | 0.595 | 7.5 | 241 | 0 | | | |
| 4.0 | 1.000 | 0.010 | 0.595 | 7.5 | 241 | 0 | × | | × |
| 4.0 | 1.500 | 0.020 | 0.893 | 7.5 | 241 | 0 | | | |
| 4.0 | 1.500 | 0.015 | 0.893 | 7.5 | 241 | 0 | | | |
| 4.0 | 1.500 | 0.010 | 0.893 | 7.5 | 241 | 0 | | | |
| 4.0 | 1.500 | 0.005 | 0.893 | 7.5 | 241 | 0 | | | |
| 4.0 | 2.000 | 0.015 | 1.190 | 7.5 | 241 | 0 | | | |
| 4.0 | 2.000 | 0.0125 | 1.190 | 7.5 | 241 | 0 | | | |
| 4.0 | 2.000 | 0.008 | 1.190 | 7.5 | 241 | 0 | × | × | × |
| 4.0 | 2.000 | 0.004 | 1.190 | 7.5 | 241 | 0 | × | | × |

**Table B3.** ABOVE test matrix. $V$ is the mean wind speed, $f_s$ the surge frequency, $A_s$ the surge amplitude, $f_r$ the reduced frequency, TSR is tip-speed ratio, RS rotor speed, $\beta$ is the blade pitch angle. The CW, AW, PIV columns indicates wether if a cross-wind, along-wind or PIV measurement of the wake was carried out.

| $V$ [m/s] | $f_s$ [Hz] | $A_s$ [m] | $f_r$ [-] | TSR [-] | RS [rpm] | $\beta$ [deg] | CW | AW | PIV |
|---|---|---|---|---|---|---|---|---|---|
| 6.0 | 0.125 | 0.125 | 0.050 | 5.5 | 265 | 12.5 | | | |
| 6.0 | 0.125 | 0.100 | 0.050 | 5.5 | 265 | 12.5 | | | |
| 6.0 | 0.125 | 0.065 | 0.050 | 5.5 | 265 | 12.5 | | | |
| 6.0 | 0.125 | 0.030 | 0.050 | 5.5 | 265 | 12.5 | | | |
| 6.0 | 0.250 | 0.125 | 0.099 | 5.5 | 265 | 12.5 | | | |
| 6.0 | 0.250 | 0.100 | 0.099 | 5.5 | 265 | 12.5 | | | |
| 6.0 | 0.250 | 0.065 | 0.099 | 5.5 | 265 | 12.5 | | | |
| 6.0 | 0.250 | 0.030 | 0.099 | 5.5 | 265 | 12.5 | | | |
| 6.0 | 0.500 | 0.100 | 0.198 | 5.5 | 265 | 12.5 | | | |
| 6.0 | 0.500 | 0.075 | 0.198 | 5.5 | 265 | 12.5 | | | |
| 6.0 | 0.500 | 0.050 | 0.198 | 5.5 | 265 | 12.5 | | | |
| 6.0 | 0.500 | 0.025 | 0.198 | 5.5 | 265 | 12.5 | | | |
| 6.0 | 0.750 | 0.065 | 0.298 | 5.5 | 265 | 12.5 | | | |
| 6.0 | 0.750 | 0.050 | 0.298 | 5.5 | 265 | 12.5 | | | |
| 6.0 | 0.750 | 0.030 | 0.298 | 5.5 | 265 | 12.5 | | | |
| 6.0 | 0.750 | 0.015 | 0.298 | 5.5 | 265 | 12.5 | | | |
| 6.0 | 1.000 | 0.070 | 0.397 | 5.5 | 265 | 12.5 | | | |
| 6.0 | 1.000 | 0.050 | 0.397 | 5.5 | 265 | 12.5 | | | |
| 6.0 | 1.000 | 0.035 | 0.397 | 5.5 | 265 | 12.5 | | | |
| 6.0 | 1.000 | 0.018 | 0.397 | 5.5 | 265 | 12.5 | | | |
| 6.0 | 1.500 | 0.030 | 0.595 | 5.5 | 265 | 12.5 | | | |
| 6.0 | 1.500 | 0.025 | 0.595 | 5.5 | 265 | 12.5 | | | |
| 6.0 | 1.500 | 0.015 | 0.595 | 5.5 | 265 | 12.5 | | | |
| 6.0 | 1.500 | 0.008 | 0.595 | 5.5 | 265 | 12.5 | | | |
| 6.0 | 2.000 | 0.018 | 0.793 | 5.5 | 265 | 12.5 | × | | |
| 6.0 | 2.000 | 0.0125 | 0.793 | 5.5 | 265 | 12.5 | × | | |
| 6.0 | 2.000 | 0.006 | 0.793 | 5.5 | 265 | 12.5 | × | | |

*Author contributions.* AF revised the experimental dataset helping to get the most recent results. IB was responsible of the full-turbine experiments and helped define the experimental methodology. RM carried out the 2D experiments and processed the respective data. MB and AZ supervised and promoted the research activity. Finally, AF prepared the manuscript of this article with contribution from all co-authors.

*Competing interests.* The authors declare that they have no conflict of interest.

*Acknowledgements.* The UNAFLOW research project has been funded by EU-EERA (European Energy Research Alliance)/IRPWIND Joint Experiment 2017.

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
