# Peer review of "UNAFLOW: a holistic wind-tunnel experiment about the aerodynamic response of floating wind turbines under imposed surge motion"

_Wind Energy Science, 2021_

## Referee Comment (RC4)

[referee-annotated manuscript omitted]

---

## Author Comment (AC4)

[Figure]

Figure 1: Spectrum of the tower-top FA shear-force for the same surge-motion (A = 0.1m, f = 0.25Hz) and no wind measured in the LIFES50+ and UNAFLOW experiments. The vertical lines mark the integer multiples of the surge-motion frequency.

---

## Author Response (AR1)

Politecnico di Milano Department of Mechanical Engineering Via La Masa 1, 20156, Milan Italy

Wind Energy Science Discussion

Date: May 07, 2021 Subject: WES-2021-12 Final Response

Dear Referees,

We would like to thank you for taking the time to review our manuscript. We are very grateful for the constructive discussion, and high-quality feedbacks you provided that we believe will improve the quality and impact of this work.

The article has been revised according to your suggestions, and you can find it at the end of this document. The attached document tries to provide a detailed answer to the comments you made in the Interactive Discussion.

On behalf of all Authors, yours sincerely,

Alessandro Fontanella

Attached documents:

- Response to Anonymous Referee #1
- Response to Anonymous Referee #2
- Manuscript changes (latexdiff)

**Response to Anonymous Referee #1**

**Dear Referee,**

Thank you for taking the time to review our manuscript and for the valuable comments you made. Below are our answers to your comments and suggestions.

- **RC1.1** The aim of the manuscript needs to be rewritten. The current aim stated in line 57 page 2 is more of a paper outline than an aim. The scientific findings that the paper is trying to uncover and the broader impact of these findings are not found.
- **AC1.1** Referee noticed that the paper aim is not clearly addressed in the manuscript introduction, that should also point out the main findings and their impact. We agree with Referee's comment, and we replaced line 48 to line 59 of the manuscript with the text below.

The unsteady response of FOWTs is still an open question. In this respect, this article presents the wind-tunnel scale-model experiment that was carried out as part of the IRPWind UNAFLOW project. The goal of the experiment is to study the aerodynamic response and wake for an FOWT subjected to large surge (i.e., translational) motion, as it normally occurs in operation. Studying these issues at small scale has some limitations because it is not possible to reproduce exactly all the physics of a full-scale system (e.g., structural response, inflow conditions). However, this disadvantage is offset by the possibility to control precisely and know better the test conditions, and to implement more measurements.

The main contributions of this work are as follows:

- a preliminary 2D experiment is performed to characterize the airfoil used for the scaled turbine blades. Unlike in previous study, knowledge of unsteady aerodynamic response of blade airfoil is leveraged to select the wind and surge-motion conditions for the full-turbine experiment. In addition, 2D data are a reliable polar-dataset that can be used to create numerical models of the experiment;
- accuracy of force measurements is improved with respect to the previous test campaigns of Bayati et al. (2016) and Bayati et al. (2017c). The flexible tower in LIFES50+ tests created issues in the measurements, making their use for code validation difficult;
- thrust force measurements from full-turbine experiments are compared to predictions of a quasi-steady rotor-disk model. This model is often relied on when building reduced-order FOWT models for control applications (e.g., Lemmer et al. (2020); Fontanella et al. (2020)) and assessing its prediction capabilities is therefore crucial for developing effective controllers. It is found the thrust force follows the quasi-steady theory for reduced frequency below 0.5;
- the wind turbine wake is measured with hot-wire probes to describe and quantify the effect of surge motion on its energy content. PIV measurements are utilized to assess the influence of surge motion on the position of tip-vortex inside the wake. The wake energy is increased in correspondence of the surge-motion frequency.

The impact this paper and the UNAFLOW experiment have on research about FOWT unsteady aerodynamics is:

- additional knowledge about the unsteady aerodynamics of an FOWT. In particular, the analysis is carried out with a system engineering vision of the problem, that considers the response of the entire floating system. Its findings may have an impact on blade design, wind turbine control, wake interaction and wind farm control.
- experimental methodology. The UNAFLOW experiment is the result of a joint effort of different research groups, some expert in numerical simulations and some in scale-model experiments. The experiment followed an integrated approach: results of numerical computations and 2D experiments were utilized to design full-turbine experiments, which results were in turn used for validation of numerical tools. Because of these aspects, the experiment can be considered among the most advanced wind tunnel test about FOWT unsteady aerodynamics to date.
- database. Differently than the previous test campaigns of Bayati et al. (2016) and Bayati et al. (2017c), the UNAFLOW experiment generated a comprehensive database that covers in a coherent manner different aspect of the unsteady aerodynamic response of an FOWT: aerodynamic coefficients of the blade airfoil, rotor-integral forces, and near-wake. The database is accessible at:

https://doi.org/10.5281/zenodo.4740005

The systematic approach of the experiment makes data especially useful for validating numerical tools. Cormier et al. (2018) utilized the UNAFLOW data to assess predictions of a BEM, a free-vortex and a fully-resolved CFD model. A second comparison with numerical tools was recently carried out by Mancini et al. (2020). The UNAFLOW dataset is currently used for the validation of numerical codes in the IEA Wind Task 30 OC6 project.

- **RC1.2** The coordinate system used in the manuscript is not clear. It is hard to tell whether "y" is the vertical or spanwise direction. I suggest a clear statement clarifying the coordinates so it is easier to follow the results' discussion. Also, the axes in the figures should be checked for the same issue (e.g.figure 14) and ensure that the coordinates are consistent throughout the manuscript.
- AC1.2 As pointed out by Referee's (actually, also Referee #2 noticed that) comment, the coordinate systems of the article were not clear. Moreover, an error in the y-axis label of Fig. 14 contributed to increase confusion about the reference frame of the PIV results. To solve this issue, we replaced Fig. 6 of the manuscript with the figure below,

that shows the principal coordinate systems of the article. The y-axis label of Fig.14 has been corrected accordingly ("y/R [-]" --> "z/R [-]").

Figure 1. Schematic of the full-turbine test setup and coordinate systems used for measurements.

- **RC1.3** Please revise the manuscript for typos. Below are some of the technical comments
- AC1.3 Page 2 Line (29-31): Unclear sentence. "Even though the importance of the aerodynamics of FOWTs is widely recognized, few are the experiments that tried to shed light into this topic".

We agree that this sentence is unclear, and we replaced it with: "To date, the aerodynamics of FOWTs was studied in a restricted number of experiments".

• Page 3 Line 69: because the flow (check).

We changed the sentence at line 69 in: "FOWTs undergo large rigid-body motions that are permitted by the high-compliance of the floating foundation and wave forcing. Consequently, the rotor of an FOWT often operates in strong unsteady-flow conditions".

Page 3 Line 82: The second condition. (which one the second condition, please clarify).

Here "second" is mistaken for "third". To improve clarity, we decide to rewrite the sentence as: "In the ABOVE condition, the TSR is lower and the collective pitch angle is increased, to get a lower power coefficient."

• Page 3 Line 83: what is the turbulence index?

Misspelled, it is turbulence intensity that was computed as the ratio between the standard-deviation of the turbulent velocity fluctuations and the mean velocity

**Response to Anonymous Referee #2**

**Dear Referee,**

Thank you for the extended and accurate feedback.

We agree with you that it is useful to make the dataset more accessible for future research. For this reason, we decided to upload it on Zenodo and we included the reference to the first version of the database in the manuscript.

We are grateful for all comments about figures and text structure. We kept them in great consideration as we are sure they can improve the paper quality.

Finally, there are some comments in the edited version of the paper we would like to answer here, because we think they are of great value for the article and our research in general.

**RC2.1** I'm not sure I would agree with this statement. If you are looking at tools to design floating wind systems, many were adapted from land-based and offshore tools directly for floating wind, with limited use for bottom-fixed. I would say rather that they were adapted from land-based tools.

Referred to: "Wind turbines and wind farms are designed and studied by means of numerical simulation tools, that are in large part developed for bottom-fixed wind turbines".

**AC2.1** Thank you for this comment. We think our sentence was inaccurate and the one you proposed is closer to what we meant. We replaced the old sentence with a short comment about the unsteady aerodynamic response of floating turbines and why it is important to assess if the current aerodynamic tools can predict it correctly.

Wind turbines and wind farms are often designed by means of engineering tools that were adapted from land-based tools. In this adaptation process, the aerodynamic model has remained almost unchanged. However, floating turbines are subjected to peculiar inflow conditions that are not present in land-based turbines. The rotor of land-based turbines undergoes small-amplitude motions associated to the tower flexible response. The motion of an FOWT rotor is in large part set by the rigid-body motion of the support platform and is in general of higher amplitude and lower frequency than in land-based turbines. The accuracy of land-based-derived aerodynamic tools in this new inflow conditions is yet to be assessed. An accurate prediction of the aerodynamic response caused by rotor motion is crucial. As said, this occurs at lower frequencies than in land-based turbines and, differently than in the latter, it causes significant interactions with the turbine controller (i.e., the aerodynamic response in FOWTs is inside the bandwidth of the turbine controller) that may lead to instability. Experiments play a crucial role in verifying whether the aerodynamic codes are accurate also for floating turbines, to get a deeper understanding of the peculiar aerodynamic phenomena that occurs when the wind turbine undergoes large motions and, based on this knowledge, to develop better simulations tools.

**RC2.2** I'm not sure I agree with this statement. There are many floating wind tests that have been done in wind/wave facilities focused on aerodynamics, but not in a good wind environment like a wind tunnel, and with little focus on wakes.

Referred to: "few are the experiments that tried to shed light into this topic".

**AC2.2** Yes, you are right, and research about wind/wave basin tests must be mentioned in the literature survey. We did that with the paragraph reported below.

In parallel with wind-tunnel experiments, a series of floating turbine model-tests was performed in different wave-basins. Among the goals of these experiment was to investigate the effect of turbine aerodynamic loads on the global response of the system. In "Experimental Comparison of Three Floating Wind Turbine Concepts" the response to wind and wave excitation of three 5MW FOWT concepts was investigated at 1/50 scale.

The blades of the turbine model were a geometrically scaled version of the NREL 5MW blade, and the aerodynamic performance (thrust force and power) of the rotor was not representative of the full-scale turbine. This was found to be a consequence of the Froude-scaled low-Reynolds wind. To cope with this issue, a new rotor was designed, and a second set of tests was carried out in "Additional wind/wave basin testing of the DeepCwind semisubmersible with a performance-matched wind turbine". This second campaign proved that wind-turbine aerodynamic loads must be reproduced correctly when assessing the global response of FOWTs in wave-basin tests.

More recent research efforts, like "Experimental observations of active blade pitch and generator control influence on floating wind turbine response", "The Triple Spar campaign: Model tests of a 10MW floating wind turbine with waves, wind and pitch control" studied the interaction between turbine-control, aerodynamic forces, and platform motions.

Overall, integrated wave-basin tests proved to be very useful in studying the coupled response of floating turbines modeling simultaneously wave excitation, wind, and turbine control.

However, reproducing the turbine aerodynamic response is hindered by the low-Reynolds number imposed by Froude-scaling ("A wind tunnel/HIL setup for integrated tests of Floating Offshore Wind Turbines") and by the quality of the wind environment ("Methodology for Wind/Wave Basin Testing of Floating Offshore Wind Turbines"). With these limitations, reproducing a realistic turbine wake is usually out of reach.

**RC2.3** What do you mean by a low uncertainty level? Was uncertainty assessed? What was improved upon from the previous Bayati campaigns which makes this one more useful?

Referred to: "Thanks to the systematic approach, the experimental data are featured by a low uncertainty level, that promotes their use as a benchmark for the development of numerical tools".

**AC2.3** Thank you for this comment that gives us the opportunity to discuss some important points about the test campaign of this article and research activities we are planning for the near future.

Uncertainty was not quantified in the present test campaign. However, quantifying uncertainty of experimental results is important to interpret them correctly. The UNAFLOW dataset is currently utilized in the OC6 project and

quantify uncertainty of datasets used for the validation of numerical tools is among the project goals. For this reason, we are currently planning a test campaign like the one discussed in this paper but dedicated to uncertainty quantification.

We included this comment in the conclusion, among the open research questions.

Concerning the last question, we can say the UNAFLOW experiment improved the upon the Bayati campaigns in the following aspects:

- The full-turbine experiment was designed based on a systematic approach. Knowledge about the aerodynamic response of the 2D airfoil was exploited to select the wind and surge-motion conditions of the fullturbine experiment.
- Improved accuracy of force measurements. The accuracy of force data in the previous test campaigns was penalized by tower flexibility that created issues in the measurements, making their use for code validation difficult.
- The turbine wake was investigated with PIV and hotwire measurements for several surge motion conditions.

The UNAFLOW experiment generated a comprehensive database that covers in a coherent manner different aspect of the unsteady aerodynamic response of an FOWT rotor: aerodynamic coefficients of the blade airfoil, rotor-integral forces, and near-wake. The database is also freely available for the community.

This comment was included in the introduction, and together with the comments we added to answer Referee #1 observations, it helps clarifying the impact of this work.

**RC2.4** What are the limitations of studying these issues at this small scale?

Referred to: "The purpose of the wind tunnel experiment was to provide a large dataset of rotor integral loads and wake measurements for several wind-turbine operating and motion conditions, selected to be realistic for a multi-megawatt FOWT. 2D sectional airfoil experiments were carried out prior to the full-turbine tests, to guide the selection of the motion conditions for the turbine scale model, and to support the creation of numerical models of the experiment".

**AC2.4** Thank you for this comment. We think it helps putting our work, and scale-model experiments in general, in perspective, underlying the advantages with respect to full-scale tests. We modified the previous text as it follows.

2D sectional airfoil experiments were carried out mainly for two reasons. Knowledge of the airfoil response was leveraged to select the surge-motion conditions of the full-turbine experiment and to provide a reliable polar-dataset that can be used to create numerical models of the experiment. Studying these issues at small scale has some limitations because it is not possible to reproduce exactly all the physics of a full-scale system (e.g., structural response, inflow conditions, ...). However, this disadvantage is offset by the possibility to control precisely and know better the test conditions, and to implement more measurements.

**RC2.5** Was the tower frequency designed to represent a scaled version of the full-scale design?

Referred to: "The maximum frequency investigated in the full-turbine experiment was limited to 2 Hz to avoid exciting the first tower fore-aft flexible mode".

And:

Could the system not be made stiffer to eliminate this issue?

Referred to: "It would be interesting to investigate surge-frequencies higher than those considered in this experiment. This is in general complex because of the risk of exciting the flexible modes of the wind turbine scale model".

**AC2.5** Thank you for these comments that helped clarifying the tower-flexibility issue.

The tower of the LIFES50+ turbine ("Wind Tunnel Wake Measurements of Floating Offshore Wind Turbines") was designed to match the 1st FA mode of the DTU 10MW (6.29 Hz at model scale) but turned out to be more compliant than desired (4.25 Hz), probably because the properties of the carbon fiber used in the production process were different than those considered in the design. The UNAFLOW turbine adopted a new tower design based on aluminum instead of carbon fiber. The new tower is stiffer (1st FA mode at 6.75 Hz).

The flexible response of the LIFES50+ tower penalized force measurements. This is exemplified by Figure 2, that compares the spectrum of the tower-top FA shear-force for the same surge-motion (A = 0.1m, f = 0.25Hz) and no wind measured in the LIFES50+ and UNAFLOW experiments. In no wind condition, the FA force is mostly due to RNA inertia and as visible, the contribution due to the tower resonant response was larger in LIFES50+ than in UNAFLOW.

It is desirable to increase the frequency of the 1st FA mode to avoid resonant excitation due to higher harmonics of the imposed surge-motion, that decrease with frequency. The turbine FA modes are set by tower stiffness and weight of the rotor-nacelle assembly. Frequency is increased by reducing the latter and increasing the former. Slight stiffness increments are possible modifying the tower design, for example using carbon fiber in place of aluminum. RNA mass is instead heavily constrained by the mass of actuators (generator and pitch), control electronics, and sensors, that are commercial components, and cannot be modified.

We commented the tower flexibility issue in the section about design of the experiment. Moreover, we explained in the conclusion that it is desirable to increase further the tower stiffness. However, this may prove to be difficult, and "numerical experiments" can assist physical tests in exploring the high frequency response.

---

## Author Response (AR2)

Politecnico di Milano
Department of Mechanical Engineering
Via La Masa 1, 20156, Milan
Italy

Wind Energy Science Discussion

Date: August 16, 2021
Subject: WES-2021-12 – manuscript needs minor revisions

Dear Athanasios,

Thank you for having reviewed the last version of our paper. We think your comments helped us improving it further.

The article has been revised according to your suggestions. On top of that, we double-checked the manuscript and took the opportunity to clarify some sentences and correct typos.

Below you can find a point-by-point reply to your comments.

On behalf of all Authors,
yours sincerely,

Alessandro Fontanella

**Point-by-point reply to comments**

Below you can find our reply (AR) to your comments (AEC).

**AEC1**    L 83: the quasi-steady theory seems valid for reduced frequency smaller than 0.5. How does this value compare with other studies? Also, how does that translate to motion frequencies of model and full scale turbines?

**AR1**    Thank you for this comment that helps us to better frame our research.
To answer the comment, we did two things. In section 2.2 we added a reference to a recent paper of C. Ferreira, currently under review, that summarizes results of previous experimental and numerical studies about the aerodynamic response of floating turbines. In this survey, results are reported as function of reduced frequency and this gives a good indication of the reduced-frequency range explored in other studies. Furthermore, we recognize that reporting results as a function of reduced frequency favors comparisons, so we decided to discard the "wake reduced velocity" (the inverse of reduced frequency) we introduced in the first manuscript, and to use reduced frequency in any part of the paper. The test matrix in the appendix now reports the reduced frequency of all tested cases.
We added a short appendix where wind tunnel results are extended to a generic 10MW floating turbine. Here, we check how reduced frequency varies as function of surge-motion frequency and wind speed and based on this information, we notice that it's reasonable to expect quasi-steady aerodynamics in response to resonant surge motion.

**AEC2**    Fig 2: I suggest to indicate with a dash vertical line the radial position of the end of the blade root (i.e. the radial location at which the circular cross-section of the blade ends).

**AR2**    Done, and we did the same in Fig. 1.

**AEC3**    L 176: It might be clearer to describe in a bit more details how the obtained 2D airfoil characteristics are used to decide on the set up for the 3D experiment.

**AR3**    We agree that it was not very clear how we use 2D data to design the 3D experiment. Hence, we made some modifications to the Introduction and Section 2.2 to better show how 2D data were used. Knowledge of airfoil polars was utilized to ensure that angle-of-attack variation due to surge motion do not cause unsteady airfoil aerodynamics. In this way it is possible to say any turbine unsteady aerodynamic behavior is due to rotor unsteadiness rather than airfoil-level unsteadiness. We added one panel to Fig. 2 to better explain this idea.

**AEC4**    L 225: Is 5m upstream of the turbine enough to measure U_infty?

**AR4**    That's correct and in fact it's an error: wind speed was measured 7.15m upstream the turbine rotor. Here the influence of rotor induction should be negligible.